# Promiscuous and multivalent interactions between Eps15 and partner protein Dab2 generate a complex interaction network

Andromachi Papagiannoula [1,2,4], Ida Marie Vedel [1,4], Kathrin Motzny[1], Maud Tengo[3], Arbesa Saiti[1] & Sigrid Milles [1,3] ✉

Clathrin-mediated endocytosis depends on complex protein interactions. Eps15 plays a key role through interactions of its three EH domains with Asn-Pro-Phe (NPF) motifs in intrinsically disordered regions (IDRs) of other endocytic proteins. Using nuclear magnetic resonance spectroscopy, we investigate the interaction between Eps15's EH domains and a highly disordered Dab2 fragment (Dab2$_{320-495}$). We find that the EH domains exhibit binding promiscuity, recognizing not only the NPF motif of Dab2 but also other phenylalanine containing motifs. This promiscuity enables interactions with Eps15's own IDR (Eps15$_{IDR}$), which lacks NPF motifs, suggesting a self-inhibitory state that promotes liquid-liquid phase separation. Despite competing for the same EH domain binding sites, Eps15$_{IDR}$ and Dab2$_{320-495}$ can bind EH123 simultaneously, forming a highly dynamic interaction network that facilitates the recruitment of Dab2$_{320-495}$ into Eps15 condensates. Our findings provide molecular insights into the competitive interactions shaping the early stages of clathrin-mediated endocytosis.

Clathrin-mediated endocytosis (CME) is the major pathway for cargo uptake into the eukaryotic cell, often with transmembrane receptors as cargoes. The important cellular process requires a complex network of interactions, which finally results in the formation and uptake of a clathrin-coated vesicle. During the early phases of the process that takes roughly one to two minutes[1], intrinsically disordered protein regions (IDRs) of the endocytic initiators play a major role[2]. While folded proteins mainly exist in one stable conformation, IDRs interconvert rapidly between many different conformations due to their relatively flat energy landscape. IDRs often recognize their folded partner proteins through small interaction regions of only a few residues, also called linear motifs[3,4]. This is also the case in CME, where IDRs of endocytic initiators interact with, for example, the major adapter protein complex AP2 or clathrin[5–9], allowing them to establish a robust multivalent interaction network which allows for rapid rearrangement of the different protein members. Eps15, together with

FCHo1/2 proteins, counts among the first proteins to arrive at the endocytic pit[10]. It has three N-terminal Eps15 homology domains (EH domains), which share the common fold of two helix-loop-helix motifs (EF-hands) spaced by a short β-sheet[11]. The EH domains are followed by a central coiled-coil domain, through which Eps15 dimerizes[12,13], and a large IDR of around 400 residues in length (Fig. 1A)[14]. Eps15 tetramers, formed by two anti-parallel Eps15 dimers, have also been observed, seemingly requiring interaction between the EH domains and the IDR from opposite dimers[13]. FCHo1/2, which also forms dimers, uses its C-terminal μ homology domain (μHD) to interact with the multiple Asp-Pro-Phe (DPF) motifs contained in Eps15's IDR[15]. The multivalent interaction network between Eps15 and FCHo1/2 enables the formation of liquid-like protein droplets, which has been proposed as a process to catalyze the early steps of CME[16]. The EH domains of Eps15, for example, can bind to Asn-Pro-Phe (NPF) motif containing proteins[17], such as Epsin1, Stonin2[18] or Dab2 (disabled homolog 2)[19]. While

[1]Leibniz-Forschungsinstitut für Molekulare Pharmakologie, Robert-Rössle-Straße 10, Berlin, Germany. [2]Freie Universität Berlin, Department of Biology, Chemistry, Pharmacy, Berlin, Germany. [3]Université Grenoble Alpes, CNRS, CEA, IBS, Grenoble, France. [4]These authors contributed equally: Andromachi Papagiannoula, Ida Marie Vedel. ✉e-mail: milles@fmp-berlin.de

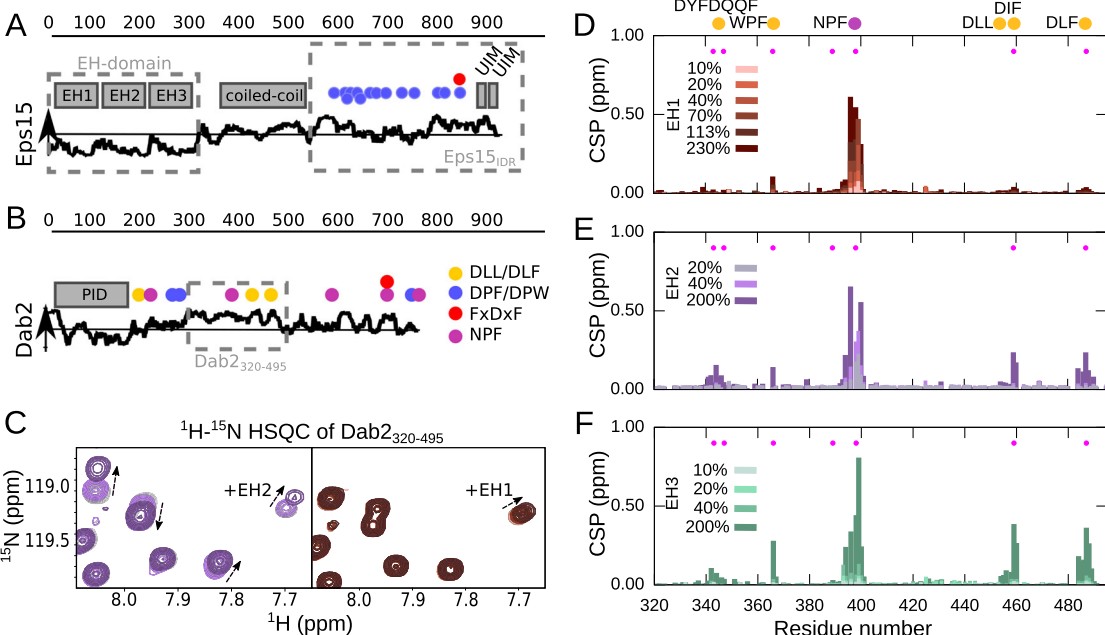

**Fig. 1 | Interaction between Dab2_{320-495} and the individual EH domains of Eps15.** Schematic illustration of (**A**) Eps15 and (**B**) Dab2. The disorder prediction (IUPred2A[23]) is shown as a black curve along the sequence. Above horizontal line: predicted disordered, Below horizontal line: predicted ordered. The folded domains are shown as gray boxes above the disorder prediction. Putative interaction motifs are shown as colored circles along the sequence (legend on the right of Dab2). DLL/DLF: clathrin binding; DPF/DPW: AP2α binding; FxDxF: AP2β binding;

NPF: EH domain binding. The stretches used in this study are boxed with gray dashed lines. **C** Zoom into a $^1H–^{15}N$ HSQC spectrum of Dab2_{320-495} in the absence and presence of EH2 (left) and EH1 (right). **D–F** CSPs of 100 µM Dab2_{320-495} in the presence versus the absence of increasing concentrations of EH1 (D), EH2 (E) and EH3 (F), respectively. Color legends are displayed in the respective plots. Filled pink circles denote positions of phenylalanines.

interactions between EH domains and both Epsin1 and Stonin2 have been characterized at molecular detail, the nature of Dab2's interaction with EH domains remains to date elusive. Dab2 has an N-terminal folded phosphotyrosine interacting domain (PID), which it uses to anchor itself to the membrane and transmembrane cargo, followed by a large IDR of around 500 residues in length, which comprises both DPF as well as NPF motifs (Fig. 1B)[20]. Both Eps15 and Dab2 belong to the class of clathrin-associated sorting proteins (CLASPs), which regulate cargo sorting during the early phases of CME and usually rely on downstream factors, such as the major adaptor protein complex AP2 or clathrin[3,21,22]. Dab2 has special properties as a CLASP since it can lead to successful CME of specific cargoes even in the absence of AP2. These cargoes, for example, integrin β1, are specifically recognized by the Dab2 PID. Importantly, binding of Eps15 to Dab2 is required for the AP2-independent internalization of these protein cargoes[19], highlighting the importance of understanding the communication between Eps15 and Dab2.

Comprehending an interaction between two proteins that comprise a significant content of IDRs is challenging. In this work, we therefore chose to use nuclear magnetic resonance (NMR) spectroscopy, which is the only experimental technique able to provide amino-acid resolution of such dynamic protein systems. Using NMR spectroscopy, we thus investigated the region of Dab2 that is predicted to be most disordered, residues 320–495 (Dab2_{320-495}), its conformational sampling, as well as its interactions with the EH domains of Eps15. All EH domains interact preferentially with the only NPF motif within Dab2_{320-495}, but EH2 and EH3 demonstrate a high level of binding promiscuity towards other phenylalanine-containing motifs. This promiscuity is maintained for the combined EH domains, comprising EH1, EH2 and EH3 (EH123), and surprisingly enables Eps15's own IDR (Eps15_{IDR}), which does not possess any NPF motifs, to bind to EH123 and thereby partially occupy the binding pockets of the different EH domains. We provide a detailed molecular characterization of

these intra-molecular interactions within Eps15, which are likely key to the liquid-liquid phase separation of Eps15 in the absence of other endocytic proteins. Our NMR results indicate that both Dab2_{320-495} and Eps15_{IDR} can bind EH123 simultaneously, but partially compete with each other. In line with this, we observe that Dab2_{320-495} is recruited into liquid-like condensates formed by full-length Eps15. Our results provide a detailed description of the multivalent and promiscuous interactions between Eps15 and Dab2, which appear to be key to a dynamic protein network that can phase separate and evolve over time, which is required in the early stages of CME.

## Results

### Dab2_{320-495} is an IDR with mild helical propensities

The around 500 residue long IDR of Dab2 comprises a central region, which is predicted by IUPred2A[23] to be significantly more disordered than the remainder of the IDR (Fig. 1B). We designed a construct encompassing this region, reaching from residue 320 to 495 (Dab2_{320-495}). This region contains 2 putative binding sites for clathrin (DLL/DLF) and one NPF motif, prone to bind to EH domains contained in Eps15. In order to assess this construct by NMR spectroscopy and describe its conformational sampling, which is fundamental to its function, we first expressed and purified the construct, and assigned the NMR backbone resonances of a $^1H$, $^{15}N$, $^{13}C$ labeled sample (Supplementary Fig. 1). We were initially not able to assign two small regions within Dab2_{320-495}, for which we designed two new stretches (Dab2_{328-360} and Dab2_{358-390}), whose $^1H–^{15}N$ heteronuclear single quantum coherence (HSQC) spectrum overlaid well with the one of Dab2_{320-495} (Supplementary Fig. 2), enabling the transfer of assignments from Dab2_{328-360} and Dab2_{358-390} to Dab2_{320-495}. As predicted, Dab2_{320-495} appears to be disordered along its entire length, testified by secondary chemical shifts (SCSs) close to zero (Supplementary Fig. 3). However, calculating a conformational ensemble on the basis of chemical shifts using a combination of the statistical coil generator

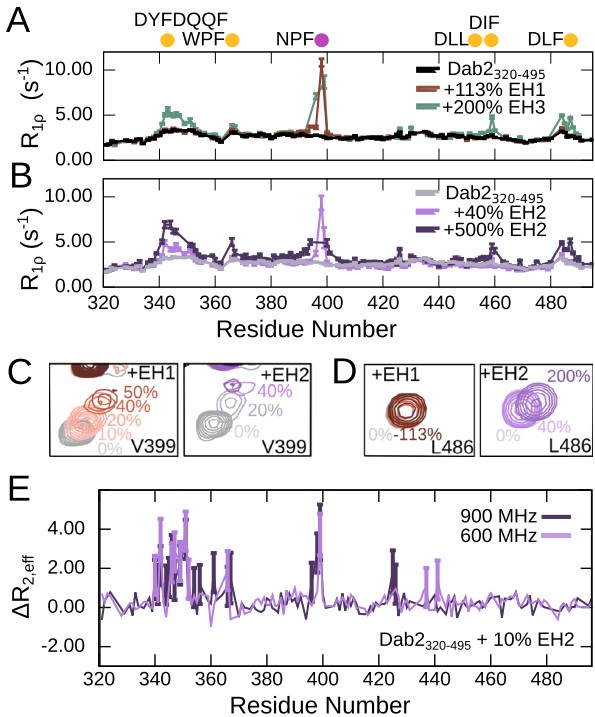

**Fig. 2 | $^{15}$N spin relaxation of Dab2$_{320-495}$ upon interaction with the EH domains.** $^{15}$N R$_{1\rho}$ spin relaxation of Dab2$_{320-495}$ (100 μM) in the absence and presence of **(A)** EH1 or EH3 and **(B)** increasing concentrations of EH2. The experiments were recorded at a $^1$H frequency of 600 MHz. The relaxation rates in **(A)** and **(B)** were derived from a fit of peak intensities against the relaxation delay. Errors of the fitted rates were derived from the experimental uncertainty. **C** Zoom into the peak corresponding to V399 from Dab2$_{320-495}$ $^1$H-$^{15}$N HSQC spectra alone and in the presence of increasing concentrations of EH1 (left) and EH2 (right). **D** Zoom into the peak corresponding to L486 from Dab2$_{320-495}$ $^1$H-$^{15}$N HSQC spectra alone and in the presence of increasing concentrations of EH1 (left) and EH2 (right). Color legends are displayed in the respective panels. **E** $\Delta$R$_{2,\,eff}$ calculated from effective R$_2$ values (R$_{2,\,eff}$) of relaxation dispersion experiments at a CPMG frequency of 31 Hz subtracted with R$_{2,\,eff}$ at a CPMG frequency of 1000 Hz recorded on a sample of 100 μM Dab2$_{320-495}$ in the presence of 10% EH2 at a $^1$H frequency of 600 MHz and 900 MHz. Error bars, propagated from the errors of R$_{2,\,eff}$ at 31 Hz and 1000 Hz determined from repeat measurements and with a minimum error of 0.5 (see Supplementary Fig. 7B for the corresponding decay curves), are shown for values that are significantly larger than 0.

flexible meccano[24] and the selection algorithm ASTEROIDS[25] revealed three regions with mild helical propensity as compared to random coil: 339–354, 454–460, and 486–490 (Supplementary Fig. 3). Around the NPF motif, no particular structural propensity could be observed. We then recorded $^{15}$N spin relaxation rates, exquisitely sensitive to the fast motions that intrinsically disordered proteins (IDPs) and IDRs undergo, in order to understand the dynamics of the intrinsically disordered chain of Dab2$_{320-495}$. In agreement with the calculated SCSs, $^{15}$N spin relaxation rates (R$_{1\rho}$, R$_1$, {$^1$H}-$^{15}$N heteronuclear Overhauser effect - hetNOE) are reminiscent of those of an intrinsically disordered protein (Supplementary Fig. 4), with increased rigidity, testified by increased R$_{1\rho}$ and hetNOE rates, apparent around the transient helix between residues 328 and 360 (helix$_N$). This helix is connected with the remainder of the disordered chain by a few amino acids of more rapid mobility. The more C-terminal transient helices are also characterized by slightly increased R$_{1\rho}$ and hetNOE rates, although less pronounced than those of helix$_N$.

## EH2 and EH3 are promiscuous binders
We then titrated the different Eps15 EH domains (EH1, EH2, or EH3) into $^{15}$N Dab2$_{320-495}$ and recorded $^1$H-$^{15}$N HSQC spectra at the different

titration steps to assess which residues in Dab2 take part in the interaction. While small chemical shift perturbations (CSPs) are observed very locally upon addition of EH1, the spectrum of Dab2$_{320-495}$ is more significantly perturbed when EH2 and EH3 are present at comparable concentrations (Fig. 1C–F and Supplementary Figs. 5, 6). Plotting the CSPs of Dab2$_{320-495}$ upon interaction with the different EH domains along its sequence (Fig. 1D–F) clearly illustrates that EH1, EH2, and EH3 have different binding patterns: EH1 is the most selective of all Eps15 EH domains, binding mainly to the NPF motif of $^{15}$N Dab2$_{320-495}$, while very small CSPs occur around residue F366 only at 200% EH1. EH2 and EH3, on the other hand, were able to interact with multiple small stretches on Dab2$_{320-495}$, revealing a high degree of binding promiscuity. This binding behavior is also mirrored by decreased peak intensities (Supplementary Fig. 6) and increased R$_{1\rho}$ rates around the respective interaction sites (Fig. 2A, B). Since a spin-lock field of 1500 Hz was used in the R$_{1\rho}$ experiments, effectively quenching contributions of intermediate exchange to the relaxation rates, the observed increases in R$_{1\rho}$ reflect the slowed rotational tumbling times of the interacting residues when in contact with the respective EH domain. Residues in between the interaction sites have R$_{1\rho}$ rates similar to those of the unbound Dab2$_{320-495}$, suggesting that the different interaction sites act independently from each other. The acquisition of R$_{1\rho}$ experiments of the different titration steps thus provides critical information on the dynamic behavior of Dab2's intrinsically disordered chain and how this is affected by binding to the folded – and much more slowly tumbling – EH domains. Even though the largest CSPs were observed around the NPF motif when EH2 and EH3 were added, also residues around DYF, WPF, DLF, and DIF motifs were in contact with those two EH domains at higher EH:Dab2 ratios. This is also reflected by the $^{15}$N R$_{1\rho}$ rates measured (Fig. 2A, B). Common to all motifs is the phenylalanine residue, suggesting an importance for this residue type in the binding to EH2 and EH3 (Fig. 1E, F). Interestingly, most of these motifs are located in the regions with increased helical propensity of Dab2$_{320-495}$ (Supplementary Fig. 3).

The Dab2$_{320-495}$ motifs were furthermore observed to bind to the EH domains with different dynamics. For example, V399, just next to the NPF motif, showed only mild peak broadening upon interaction with EH1, justified by the slowed rotational tumbling times when in complex. Upon interaction with EH2, however, significantly stronger broadening was observed (Fig. 2C). The other interacting regions show very little broadening and are mainly characterized by CSPs. One example is the residue L486, which does not interact with EH1 and shows essentially no line broadening combined with small CSPs in the presence of EH2 (Fig. 2D). These differential behaviors prompted us to record Carr-Purcell-Meiboom-Gill (CPMG) relaxation dispersion of Dab2$_{320-495}$ to investigate the possibility of intermediate exchange in the microsecond to millisecond regime between the bound and unbound states in the presence of EH2 (Fig. 2E). These experiments confirm that the NPF motif binds EH2 in intermediate exchange on the NMR chemical shift timescale. Surprisingly, many residues throughout the helix$_N$ also displayed dynamics on the intermediate timescale upon binding. The remaining motifs, DIF and DLF, were concluded to bind in fast exchange.

In order to quantify the affinity of the interaction between Dab2$_{320-495}$ and the EH domains, we attempted to extract dissociation constants (K$_D$) for the different Dab2$_{320-495}$ interaction motifs. Even though fitting of CSPs as a function of EH domain concentration was difficult due to the low affinities and peak broadening (especially around the NPF motif upon interaction with EH2), they can be estimated to be in the high micromolar to millimolar range. The highest affinity of ~340 μM was determined between residue G400 and EH2– this residue is close to the NPF motif, which broadens too rapidly in the titration to determine a K$_D$. EH1 interacts with the NPF motif (G400) with an affinity similar to that of EH2 (~350 μM), while the affinity of EH3 was estimated to only ~1.4 mM. The other, promiscuous binding

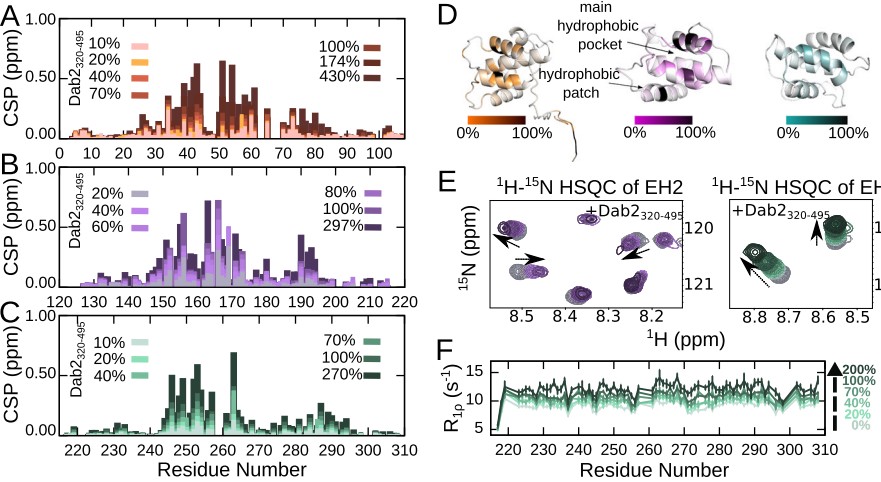

**Fig. 3 | Interaction between the individual EH domains of Eps15 and Dab2$_{320-495}$.** **A–C** CSPs of 100 µM EH1 (**A**), EH2 (**B**) and EH3 (**C**), respectively, in the absence versus the presence of increasing amounts of Dab2$_{320-495}$. Color legends are displayed in the respective plots. **D** Structures of EH1 (AlphaFold2[34]), EH2 (PDB 1FF1[31]) and EH3 (PDB 1C07[32]) with the CSP at 100% Dab2$_{320-495}$ mapped onto them. The linear gradient starts from 0% (white), which represents no CSP, to the 100% (black), which is 0.75 ppm (the biggest CSP observed between EH2 and

Dab2$_{320-495}$). **E** Zoom into a $^1$H-$^{15}$N TROSY-HSQC spectrum of EH2 (left) and EH3 (right) alone and in the presence of Dab2$_{320-495}$. **F** $^{15}$N R$_{1\rho}$ spin relaxation of EH3 (100 µM) in the absence and presence of Dab2$_{320-495}$ at a $^1$H frequency of 600 MHz. Color legends are indicated in the figures. The relaxation rates were derived from a fit of peak intensities against the relaxation delay. Errors of the fitted rates were derived from the experimental uncertainty.

sites, interact even more weakly (Supplementary Fig. 7A and Supplementary Table 1). Analysis of the CPMG relaxation dispersion curves of Dab2$_{320-495}$ upon interaction with EH2 using a two-site exchange model allowed us to extract an exchange rate of $149 \pm 13$ s$^{-1}$ and a percentage of bound Dab2$_{320-495}$ of $3.3 \pm 0.3\%$ for protein concentrations of 100 µM Dab2$_{320-495}$ and 10 µM EH2 (Supplementary Table 2). This results in a K$_D$ value of 196 µM, which is in a similar range as the affinity between Dab2$_{320-495}$ and EH2 determined by CSPs. The dispersion in the helix$_N$ and around the NPF motif were fit together in the fits of the CPMG experiments (Supplementary Fig. 7B).

### Dab2$_{320-495}$ binds to the hydrophobic pocket of EH domains
We wondered whether differences in the EH domains could explain the distinct interactions observed with the different motifs of Dab2$_{320-495}$. We thus set out to determine the binding pockets on each of the EH domains that engage in interaction with Dab2$_{320-495}$. Even though NMR assignments of the three EH domains exist[26–28], we assigned all three EH domains under our experimental conditions (Supplementary Figs. 8–10). From the carbon SCSs, we calculated secondary structure propensities (SSPs)[29], which are in agreement with previously determined NMR structures of the proteins (Supplementary Fig. 11)[30–32]. Additionally, $^{15}$N R$_1$ and R$_{1\rho}$ relaxation rates reflect small folded domains with flexible N- and C-termini, as expected (Supplementary Fig. 12). We then titrated Dab2$_{320-495}$ into the different $^{15}$N labeled EH domains and recorded $^1$H-$^{15}$N Transverse relaxation-optimized spectroscopy (TROSY) HSQC spectra of the EH domains at the different titration steps. We plotted the respective CSPs along the sequence of the individual EH domains (Fig. 3A–C) and then colored the structures of the three EH domains relative to the magnitude of the observed CSPs (Fig. 3D). Zooms into the spectra of EH2 and EH3 illustrate the spectral changes upon Dab2$_{320-495}$ binding (Fig. 3E). The largest CSPs occur around the previously identified binding pockets[18,31] and are comparable between all three EH domains (Fig. 3D). An additional binding region, a hydrophobic patch previously shown to bind a second NPF motif in Stonin2[18], is clearly visible in EH2. The region seems to participate in binding also within EH1 and EH3, albeit in a less pronounced way (Fig. 3A-D, region around residues 185–195 in EH2). R$_{1\rho}$ relaxation rates systematically increase throughout all residues when

Dab2$_{320-495}$ is titrated into the $^{15}$N EH domains, likely reflecting mildly increased tumbling times resulting from the motional drag that the bound IDR exerts (Fig. 3F)[33].

### EH123 shows increased binding promiscuity
In order to better reflect the physiological scenario, where the different EH domains in Eps15 are not isolated, but expressed in row, we investigated the interaction between Dab2$_{320-495}$ and the full EH-domain containing region (EH1-EH2-EH3, also called EH123) by titrating increasing concentrations of EH123 into $^{15}$N Dab2$_{320-495}$. As expected from the interactions of the individual EH domains, both CSPs and R$_{1\rho}$ rates reveal that EH123 is able to interact with the same regions of Dab2$_{320-495}$ as observed for EH2 and EH3 (Fig. 4A, B). Interestingly, the R$_{1\rho}$ relaxation rates of Dab2$_{320-495}$ in the presence of EH123 show increased relaxation not only of the residues of the five Dab2$_{320-495}$ binding regions, but the residues located in between the binding motifs also show mildly increased rates. This is particularly evident for the residues located between the first three binding motifs (DYFDQQF, WPF and NPF) and suggests an overall stiffening of the chain, potentially due to binding of the motifs to different EH domains within the same EH123 molecule.

While CSPs around the NPF motif remain the largest upon interaction with EH123, the difference between the interaction of EH123 with the NPF motif and the other motifs appears smaller than for interaction with the individual EH domains. Indeed, R$_{1\rho}$ rates increase much more significantly around these promiscuous sites, albeit part of the increase should be attributed to the larger size of EH123 relative to the individual domains. Nonetheless, binding affinities approximated from the CSPs for EH123 binding are significantly higher than those of the individual EH domains, ranging from around 100 to around 400 µM (Supplementary Fig. 7A and Supplementary Table 1). This suggests that once one EH domain within EH123 is bound to Dab2$_{320-495}$, the two other EH domains of the same molecule remain available for (preferential) binding to other motifs, an effect reminiscent of avidity, effectively increasing the total level of binding promiscuity. A much higher concentration of individual EH domains (3x as many) is needed to partially recover the chemical shift differences and

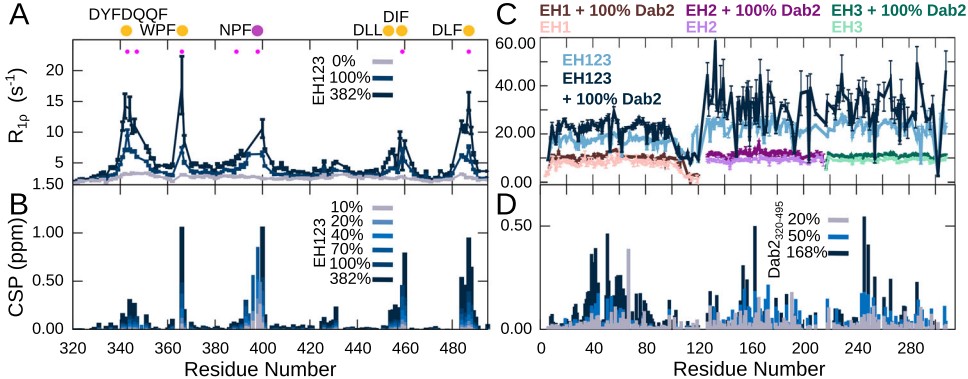

**Fig. 4 | Interaction between EH123 and Dab2$_{320-495}$. A** $^{15}$N R$_{1\rho}$ spin relaxation of Dab2$_{320-495}$ (100 μM) in the absence and presence of increasing concentrations of EH123 at a $^{1}$H frequency of 600 MHz. **B** CSPs between the $^{1}$H-$^{15}$N HSQC of $^{15}$N Dab2$_{320-495}$ in the absence and presence of increasing amounts of EH123. The filled pink dots represent the phenylalanines present in the Dab2$_{320-495}$. **C** $^{15}$N R$_{1\rho}$ spin relaxation of the individual $^{15}$N EH domains (EH1, EH2, EH3) as well as EH123 in the absence and presence of 100% Dab2$_{320-495}$ at a $^{1}$H frequency of 600 MHz. **D** CSPs between the $^{1}$H-$^{15}$N TROSY-HSQC of $^{15}$N EH123 in the absence and presence of increasing amounts of the Dab2$_{320-495}$. The color legends are indicated in the respective figure panels. The relaxation rates in (**A**) and (**C**) were derived from a fit of peak intensities against the relaxation delay. Errors of the fitted rates were derived from the experimental uncertainty.

R$_{1\rho}$ rates of Dab2$_{320-495}$ in the presence of EH123 (Supplementary Fig. 13).

**All EH domains within EH123 engage in interaction with Dab2**
To investigate whether Dab2$_{320-495}$ binds to the same sites on the EH domains when they are expressed in row rather than independently, we also aimed to characterize the interaction between Dab2$_{320-495}$ and EH123 from the side of EH123. We therefore produced $^{15}$N labeled EH123 and recorded a $^{1}$H-$^{15}$N TROSY-HSQC spectrum. If the different EH domains were unaffected by being connected to each other, we would expect this spectrum to overlay perfectly with those of the individual EH domains. However, small CSPs were observed for all three EH domains, and the peaks of EH2 and EH3 appeared broadened (Supplementary Fig. 14), revealing that each EH domain is affected by the presence of the two additional domains, potentially due to intramolecular interactions. Despite the changes in the spectrum of EH123 compared to those of EH1, EH2, and EH3, the assignments from the individual domains could be fully transferred to the combined construct. In order to examine the broadening of the peaks originating from EH2 and EH3, we recorded $^{15}$N spin relaxation of EH123 (R$_{1\rho}$). All R$_{1\rho}$ rates of EH123 were significantly increased as compared to the individual EH domains, which was expected for this larger protein construct (Fig. 4C and Supplementary Fig. 14C). The rates of EH2 and EH3, however, were significantly higher overall than those from EH1. Indeed, the increased R$_{1\rho}$ rates were similar for EH2 and EH3, suggesting that EH2 and EH3 tumble together as one entity, while EH1 can move independently within EH123. To further investigate this behavior, we decided to address the very fast, picosecond motions that are normally present in dynamic linkers between folded domains by measuring a hetNOE experiment with EH123. The recorded hetNOEs show quite similar values all along EH123, reflecting the similar fold and internal dynamics of the individual domains (Supplementary Fig. 14C). Only the linker between EH1 and EH2 displays much lower hetNOE values, demonstrating that this linker is undergoing much more rapid motion than the rest of the protein construct, also compared to the linker connecting EH2 and EH3. The hetNOE data are thus in very good agreement with EH2 and EH3 tumbling together within the EH123 construct.

We then added Dab2$_{320-495}$ to $^{15}$N labeled EH123 to assess whether EH123 bound differently to Dab2 than the individual EH domains. We observed CSPs for all three domains, similar to those of the individual EH domains when interacting with Dab2$_{320-495}$ (Fig. 4D), suggesting that the binding sites on all EH domains are also available within EH123,

despite the fact that EH2 and EH3 tumble together within EH123. R$_{1\rho}$ rates increased throughout EH123 when Dab2$_{320-495}$ was added (Fig. 4C), and the rates of EH1 remained lower than those of EH2 and EH3, suggesting that the overall conformation of the EH domains within EH123 is preserved upon binding of Dab2$_{320-495}$.

**Eps15's own IDR interacts with EH123**
The remarkable binding promiscuity towards phenylalanine-containing motifs observed between EH123 and Dab2$_{320-495}$, prompted us to investigate whether EH123 could also interact with Eps15's own IDR (Eps15$_{IDR}$). Eps15$_{IDR}$ does not contain any canonical NPF motifs, but contains 14 DPF motifs (Fig. 1A), which have previously been suggested to interact with Eps15's own EH domains[16]. When we added equimolar amounts of Eps15$_{IDR}$ to $^{15}$N EH123, CSPs were observed throughout the $^{1}$H–$^{15}$N TROSY-HSQC spectrum of EH123, confirming the interaction and further suggesting that all EH domains bind to Eps15$_{IDR}$ (Fig. 5A and Supplementary Fig. 15). Upon interaction, most peaks of EH2 and EH3 were severely broadened, likely because of the motional drag the long Eps15$_{IDR}$ exerts on EH123. As a consequence, many of the signals were too weak to calculate CSPs and extract relaxation rates for EH2 and EH3 within EH123 upon interaction with Eps15$_{IDR}$ (Fig. 5A). However, the CSPs originating from EH1 within EH123 were all smaller than those in presence of the same amount of Dab2$_{320-495}$, indicating that EH1 interacts less well with Eps15$_{IDR}$ than with Dab2$_{320-495}$, in agreement with EH1 being the least promiscuous of the domains, binding more selectively to NPF motifs than EH2 and EH3 (Fig. 5A).

**EH123 binds phenylalanine-containing motifs in Eps15$_{IDR}$**
In order to further characterize the interaction between Eps15$_{IDR}$ and EH123, we assigned the backbone resonances of $^{1}$H, $^{15}$N and $^{13}$C labeled Eps15$_{IDR}$. To circumvent the spectral overlap of the long Eps15$_{IDR}$, we designed four overlapping smaller stretches (Eps15$_{IDR\ 481-581}$, Eps15$_{IDR\ 569-671}$, Eps15$_{IDR\ 648-780}$, and Eps15$_{IDR\ 761-896}$). The $^{1}$H-$^{15}$N HSQC spectra of these stretches overlayed nicely with the spectrum of the full Eps15$_{IDR}$ (Supplementary Fig. 16), allowing the transfer of $^{1}$H and $^{15}$N resonance assignments between the spectra. We assigned 242 out of 393 resonances distributed across the IDR except for the N-terminal residues 481-521 to which no resonances could be assigned (Supplementary Fig. 17). This is likely because these residues (481–504) are part of the coiled-coil domain as predicted by AlphaFold2[34], giving rise to weak peaks in the 3D assignment spectra. Low R$_1$ and R$_{1\rho}$ relaxation rates and SCSs around 0 for both Eps15$_{IDR}$ and the smaller stretches

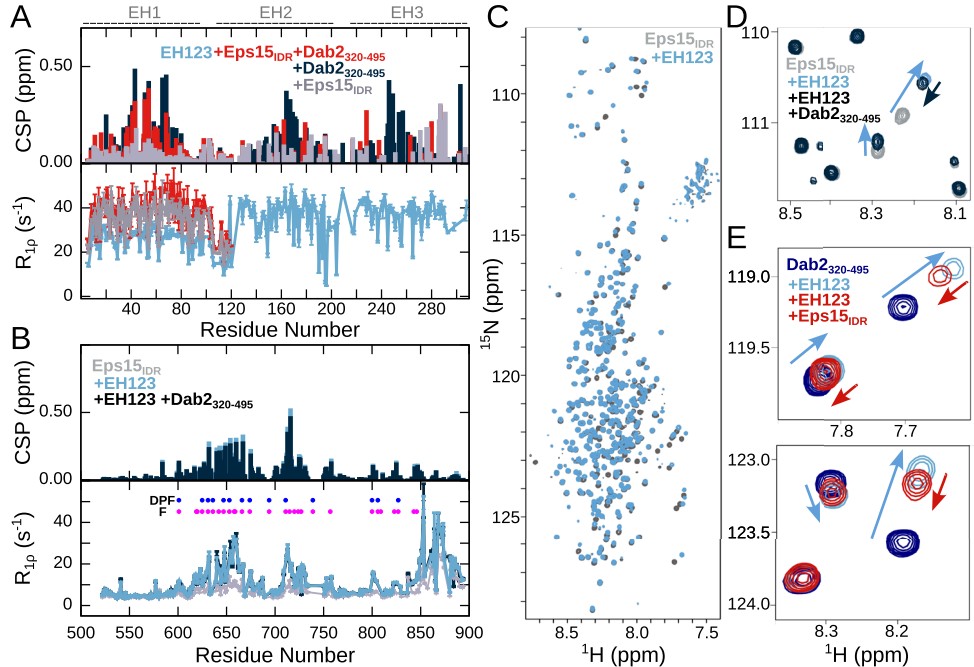

**Fig. 5 | Competitive binding of Dab2$_{320-495}$ and Eps15$_{IDR}$ to EH123. A** CSPs calculated between the $^1$H–$^{15}$N TROSY-HSQC of 200 μM $^{15}$N EH123 in the absence and presence of 200 μM Dab2$_{320-495}$ and/or 200 μM Eps15$_{IDR}$. Below are $^{15}$N R$_{1\rho}$ spin relaxation rates of $^{15}$N EH123 in the absence and presence of 100% Eps15$_{IDR}$ and both 100% Dab2$_{320-495}$ and Eps15$_{IDR}$ at a $^1$H frequency of 1200 MHz. Only the R$_{1\rho}$ rates within EH1 in the presence of Eps15$_{IDR}$ and Eps15$_{IDR}$ with Dab2$_{320-495}$ are shown as the peaks of EH2 and EH3 were severely broadened. The parts of EH123 corresponding to EH1, EH2, and EH3 are illustrated above the plots. **B** CSPs calculated between the $^1$H–$^{15}$N HSQC of 100 μM $^{15}$N Eps15$_{IDR}$ in the absence and presence of 100 μM EH123 and 100 μM EH123 + 100 μM Dab2$_{320-495}$, and $^{15}$N R$_{1\rho}$ spin relaxation rates

of $^{15}$N Eps15$_{IDR}$ in the absence and presence of 100% EH123 and 100% EH123 + 100% Dab2$_{320-495}$ at a $^1$H frequency of 1200 MHz. **C** $^1$H–$^{15}$N HSQC spectra of 100 μM $^{15}$N Eps15$_{IDR}$ alone and in the presence of 100 μM EH123. **D** Zoom into $^1$H-$^{15}$N HSQC spectrum of 100 μM $^{15}$N Eps15$_{IDR}$ alone and in the presence of 100 μM EH123 and 100 μM EH123 + 100 μM Dab2$_{320-495}$. **E** Zoom into $^1$H-$^{15}$N HSQC spectrum of 100 μM $^{15}$N Dab2$_{320-495}$ alone and in the presence of 100 μM EH123 and 100 μM EH123 + 100 μM Eps15$_{IDR}$. Color codes are denoted in the respective panels. The relaxation rates in (**A**) and (**B**) (lower panels) were derived from a fit of peak intensities against the relaxation delay. Errors of the fitted rates were derived from the experimental uncertainty.

agree well with a disordered protein (Supplementary Fig. 18). The C-terminal part however, (-residues 850–885) has increased R$_{1\rho}$ rates and SSP values close to 1, revealing a stable alpha helix. This alpha helix, which is also predicted by AlphaFold2[34], contains the two ubiquitin interaction motifs (UIMs) present in Eps15[35,36]. In general, the R$_{1\rho}$ relaxation rates are slightly larger within Eps15$_{IDR}$ compared to the smaller stretches, potentially resulting from small hydrophobic clusters in these regions leading to transient self-interactions within the chain—an effect that has been observed for transiently folded elements in other IDPs[37].

We next recorded $^1$H-$^{15}$N HSQC spectra of $^{15}$N Eps15$_{IDR}$ with equimolar amounts of unlabeled EH123 in order to pinpoint the exact interaction regions on Eps15$_{IDR}$. R$_{1\rho}$ rates, CSPs, and intensity ratios compared to Eps15$_{IDR}$ on its own revealed two relatively large interaction regions (Fig. 5B, C and Supplementary Fig. 19). These regions, 620–680 and 700–730, are characterized by a high density of phenylalanines. While the first region is enriched in DPF motifs, which have previously been proposed to bind to EH123, the second region is characterized by fewer DPF motifs and contains multiple other small linear motifs with phenylalanine, underlining the promiscuity of the EH domains. Moreover, small CSPs and increases in R$_{1\rho}$ rates are observed around three smaller regions, indicating additional interaction sites for EH123: residues -800–809 and -820–827 containing DPF(s), and residues -843–846 containing phenylalanines. While increases in R$_{1\rho}$ relaxation rates and decreases in intensity ratios in the C-terminal helix are also observed (Fig. 5B and Supplementary Fig. 19), the small CSPs accompanying these changes suggest that this may be through binding to the first helical residues of hydrophobic

nature (843–846), which then affects the tumbling time of the entire alpha helical element.

## Dab2$_{320-495}$ and Eps15$_{IDR}$ partially compete for EH123 binding

The interaction observed between EH123 and Eps15$_{IDR}$ suggests that intra-molecular interactions within Eps15 may occupy Dab2 binding sites on the EH domains in the native context. We therefore conducted a competition experiment, acquiring a spectrum of $^{15}$N EH123 with equimolar amounts of both Eps15$_{IDR}$ and Dab2$_{320-495}$. The resulting CSPs of EH123 showed that in the presence of both Dab2$_{320-495}$ and Eps15$_{IDR}$, the CSPs of EH1 were similar to those in the presence of Dab2 only. R$_{1\rho}$ rates around EH1 were increased mildly as compared to binding of Eps15$_{IDR}$ or Dab2$_{320-495}$ alone to EH123, while the peaks of EH2 and EH3 were broadened severely, also as observed upon Eps15$_{IDR}$ interaction (Fig. 5A and see Supplementary Fig. 15B). While it is difficult to disentangle the individual contributions of Dab2$_{320-495}$ and Eps15$_{IDR}$ binding to the spectral changes, this suggest that both IDRs may bind EH123 at the same time. To further investigate this, we recorded a $^1$H-$^{15}$N HSQC spectrum of $^{15}$N Eps15$_{IDR}$ and added first EH123 and then Dab2$_{320-495}$. While most peaks were unaffected by the addition of Dab2$_{320-495}$, some peaks shifted very slightly back towards the unbound state of Eps15$_{IDR}$ (Fig. 5B, D and Supplementary Fig. 19). This suggests that binding between Eps15$_{IDR}$ and EH123 is largely unaffected by the presence of Dab2$_{320-495}$. We then conducted the same competition experiment, but this time using $^{15}$N Dab2$_{320-495}$. Again, when adding equimolar amounts of Eps15$_{IDR}$ to $^{15}$N Dab2$_{320-495}$ bound to EH123, the spectrum of Dab2$_{320-495}$ was only slightly affected,

with peaks from bound $Dab2_{320-495}$ moving towards the unbound form (Fig. 5E and Supplementary Fig. 20), likely due to competition with $Eps15_{IDR}$. To rule out that any of the observed spectral changes could originate from a potential interaction between $Dab2_{320-495}$ and $Eps15_{IDR}$, we recorded a $^1H$-$^{15}N$ HSQC spectrum of $^{15}N$ $Dab2_{320-495}$ with $Eps15_{IDR}$, confirming that the two IDRs do not interact (Supplementary Fig. 21). Taken together, the competition experiments conducted from the side of each binding partner (EH123, $Eps15_{IDR}$, and $Dab2_{320-495}$) suggest that both $Eps15_{IDR}$ and $Dab2_{320-495}$ can bind to EH123 at the same time, even though they are competing for the same EH domain interaction sites. This could be possible due to a fast on and off rate of the low-affinity interaction motifs, provided binding sites in both $Dab2_{320-495}$ and $Eps15_{IDR}$ bind with similar affinities, thereby creating a complex and dynamic interaction network allowing both IDRs to bind EH123.

### $Dab2_{320-495}$ enters into Eps15 protein condensates

Eps15 has recently been shown to form liquid-like droplets, and interactions between the EH domains and $Eps15_{IDR}$ were proposed to contribute to droplet formation[16]. The seemingly simultaneous interactions of $Eps15_{IDR}$ and $Dab2_{320-495}$ with EH123 observed here made us question whether $Dab2_{320-495}$ could be recruited to Eps15 condensates. We therefore expressed and purified full-length Eps15, from here on just called Eps15, and labeled its cysteines with the fluorophore AZDye594. In order to fluorescently label $Dab2_{320-495}$, which does not contain any cysteines, we created a single cysteine mutant (S328C $Dab2_{320-495}$), and labeled it with AZDye488. We first examined if we could achieve liquid-like droplet formation of Eps15 under our experimental conditions. Indeed, upon addition of 3% PEG8000, we observed droplet formation starting from a protein concentration of as little as 0.5 μM (Fig. 6A). Droplets increased in size with increasing Eps15 concentrations (Fig. 6A, C and Supplementary Fig. 22A). Unlike Eps15, $Dab2_{320-495}$ was not able to form any droplets on its own (Supplementary Fig. 23B). When $Dab2_{320-495}$ was added to Eps15 droplets, it entered readily and clearly enriched in the Eps15 condensed phase (Fig. 6B). While the presence of $Dab2_{320-495}$ did not majorly alter the size or number of the droplets overall, we did not manage to observe phase separation at the lowest tested concentration of 0.5 μM Eps15 when 7 μM $Dab2_{320-495}$ was present in addition (Fig. 6B, C and Supplementary Fig. 22). We were thus wondering, how $Dab2_{320-495}$ concentration affected Eps15 condensation. Therefore, we added increasing concentrations of $Dab2_{320-495}$ (0.5 to 10 μM) to droplets formed by 7 μM Eps15—a concentration at which many and relatively large droplets can be observed. $Dab2_{320-495}$ entered into the droplets at all concentrations added, with the intensity of $Dab2_{320-495}$ inside the droplets increasing with higher $Dab2_{320-495}$ concentration (Fig. 6D, Supplementary Fig. 22). No major effects on condensate size or number of condensates were observed (Fig. 6E and Supplementary Fig. 22), and even 20 times molar excess of $Dab2_{320-495}$ did not seem to majorly alter the condensates (Supplementary Fig. 23). Since droplets made in the presence of smaller concentrations of $Dab2_{320-495}$ were significantly less bright than those made in the presence of higher $Dab2_{320-495}$ concentrations, we also analyzed the partition coefficient of both proteins into the droplets. While Eps15 partitioned better into droplets with increasing Eps15 concentration, its partitioning was less dependent on $Dab2_{320-495}$ concentration (Supplementary Fig. 22C, D). Partitioning of $Dab2_{320-495}$ into the droplets depended on both its own concentration in the sample, as well as the concentration of Eps15 (Supplementary Figs. 22C, D). Even though the interactions between $Dab2_{320-495}$ and EH123 are rather weak, recruitment into Eps15 droplets is specific, illustrated by a fluorescently labeled DNA, which is not recruited into Eps15 droplets (Supplementary Fig. 23E). This indicates, in line with our NMR results, that the inter-molecular Eps15 interactions responsible for droplet formation can be maintained in the presence of $Dab2_{320-495}$, while Eps15 also remains available for interaction with Dab2.

## Discussion

The dynamic network formed by intrinsically disordered regions (IDRs) and folded proteins interacting with short linear motifs within these IDRs, is key to a number of biological processes[4,38]. Clathrin-mediated endocytosis is a prime example of a process that comprises many of those short linear motifs[2,21], leading to the formation of a complex interaction network. Known examples are the α and β2 appendage domains of the major adapter protein AP2 that interact with DPF/DPW and FxDxF motifs (x = any amino acid)[3], the clathrin heavy chain terminal domain that interacts with DLL/DLF motifs[7,8] or the EH domains found in Eps15, which interact with NPF motifs of diverse IDR partners[18]. Although extensive studies have identified these consensus sequences, it has meanwhile become clear that additional binding sites are yet to be discovered and that this requires a strategy by which focus lies on studying long endocytic IDRs. This has recently allowed identifying a particularly long interaction site between the neuronal AP180 and the AP2β2 appendage domain[7]. In addition, many endocytic IDRs seem to bind to the same partners[39] and understanding how the different binding modes synergize or compete with each other is crucial for a molecular comprehension of clathrin-mediated endocytosis.

In this study, we shed light onto those partially competitive interactions by investigating the binding of a large intrinsically disordered region stemming from the CLASP Dab2 ($Dab2_{320-495}$) and its interactions with Eps15 EH domains, as well as its competition for binding with Eps15's own IDR ($Eps15_{IDR}$). In line with the previous literature and extensive peptide screens to assess binding patterns of Eps15 EH domains[40], all three EH domains of Eps15 (EH1, EH2 and EH3) prefer to bind to the only NPF motif within the sequence of $Dab2_{320-495}$ (Fig. 1, 2). In the presence of EH2, significant exchange in the microsecond to millisecond time scale is observed around the NPF motif and an N-terminal transient helix in $Dab2_{320-495}$ ($helix_N$), allowing to estimate a dissociation constant between $Dab2_{320-495}$ (NPF motif and $helix_N$) and EH2 on the order of hundreds of micromolar (Fig. 2, Supplementary Fig. 7B and Supplementary Table 2). These affinities are in agreement with affinities of other small linear motif interactions determined in the context of clathrin-mediated endocytosis[7–9] and also with those determined from chemical shift perturbations between $Dab2_{320-495}$ and EH domains (Supplementary Fig. 7A and Supplementary Table 1). However, they are orders of magnitude weaker than those observed between Stonin2 and EH2 ($K_D$ in the nanomolar range)[18]. Although the EH domains bind preferentially to the NPF motif of $Dab2_{320-495}$, as it is expected from the literature[18,31,32,40], the atomic resolution provided by our NMR experiments also points toward additional, slightly weaker binding sites including DYF, WPF, DLF, and DIF motifs (Fig. 1). Such sites will certainly play a role in the crowded environment of the endocytic pit, and these motifs further testify to an extremely promiscuous binding behavior of EH2 and EH3. This is in line with the observation that other EH domains, such as the EH domain of EHD1 or of POB-1 can bind to the motif xPF[41,42]. By investigating the EH domain interactions with small linear motifs from the side of a bona fide IDR, we can conclude that the promiscuity of EH2 and EH3 is even larger than previously anticipated, binding to most phenylalanine containing motifs (Fig. 7A). It is remarkable that, overall, the dissociation constants determined for binding between EH domains and the NPF motif in $Dab2_{320-495}$ are not extremely different to those between EH domains and non-NPF binding sites (-1.5 mM versus up to -4 mM for EH3 for example, -0.3 mM versus up to -10 mM for EH2), suggesting that promiscuity can indeed play an important role physiologically.

Even though some proteins contain individual EH domains, this is not the case for Eps15, which contains EH1, EH2 and EH3 sequentially at

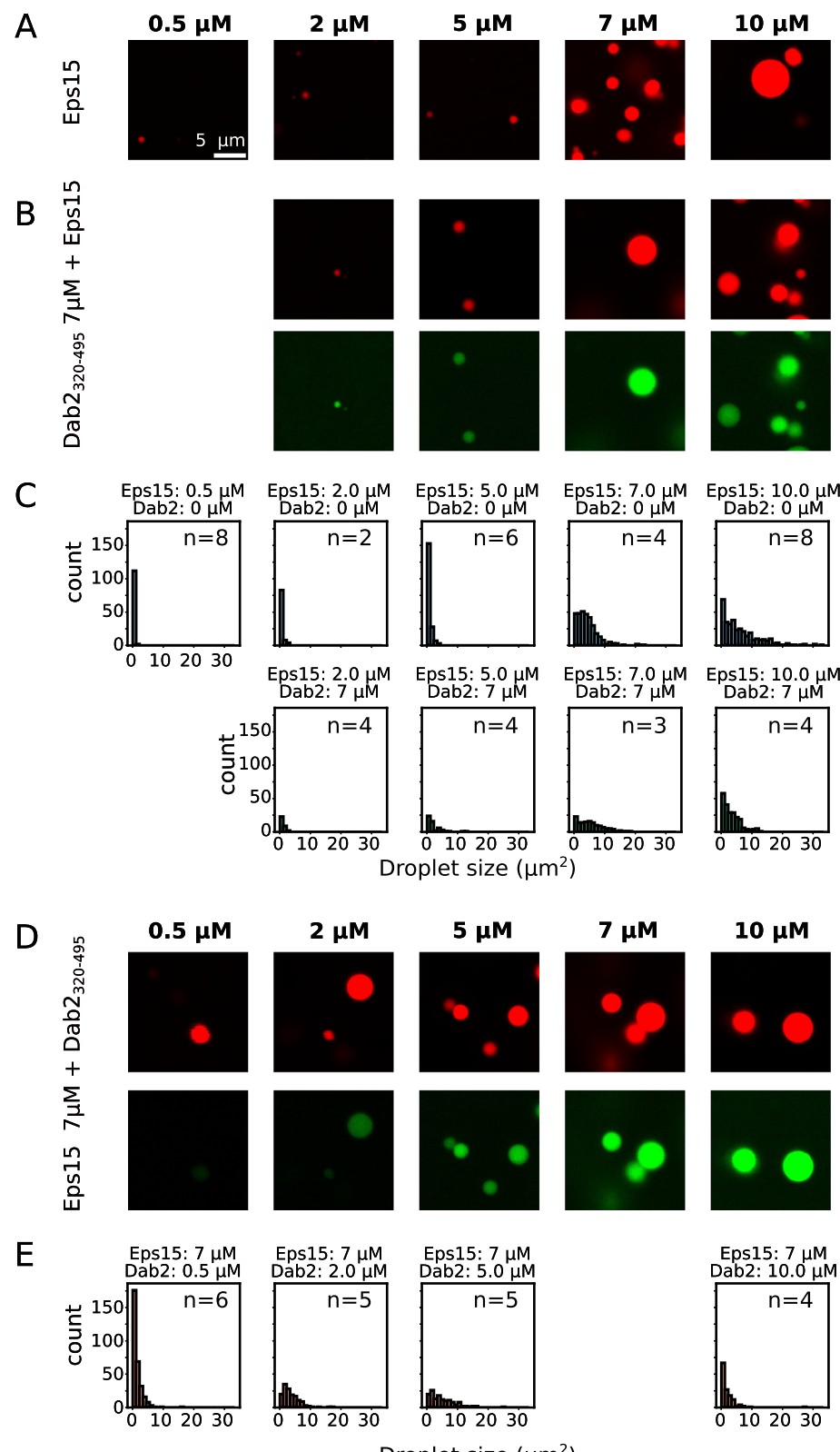

its N-terminus. Unexpectedly, we observe that EH2 and EH3 move together as one entity in a construct comprising all three EH domains (EH123, Fig. 4C), suggesting a potential interaction between EH2 and EH3. EH1, on the other hand, tumbles independently from the other two EH domains. Nonetheless, EH123 makes all EH domain binding sites available for interaction with Dab2$_{320-495}$. From the side of Dab2$_{320-495}$, binding to EH123 is strikingly different than binding to the individual EH domains. While the same NPF and promiscuous motif interactions maintain, the non-NPF interactions seem to gain importance, leading to affinities between the different Dab2$_{320-495}$ motifs in the hundreds of μM range for both NPF and non-NPF motifs (Fig. 4, Supplementary Figs. 7A, 13 and Supplementary Table 1). This could be due to avidity effects of the three EH domains in close spatial proximity compared to the individual domains.

**Fig. 6 | Dab2$_{320-495}$ is recruited into Eps15 droplets. A** Representative droplets imaged from Eps15 alone at different concentrations (0.5 to 10 μM, indicated above). **B** Eps15 droplets formed under the same conditions in the presence of 7 μM Dab2$_{320-495}$. **C** Droplet sizes plotted in a histogram for different concentrations of Eps15 in the presence and absence of 7 μM Dab2$_{320-495}$, corresponding to the microscopy images in (**A**) and (**B**). **D** Eps15 droplets formed with 7 μM protein, to which different concentrations of Dab2$_{320-495}$ (0.5 to 10 μM, indicated above) was added. **E** Droplet sizes plotted in a histogram for different concentrations of

Dab2$_{320-495}$ in the presence and absence of 7 μM Eps15, corresponding to the microscopy images in (**D**). 10% of the protein used in the microscopy experiments was fluorescently labeled. Eps15 was labeled using AZDye594 (displayed in red). Dab2$_{320-495}$ S328C was labeled using AZDye488 (displayed in green). The scale bar is 5 μm and applies to all images in the figure. Images were acquired by confocal microscopy. The number of images used for the histograms is indicated as 'n = x' in each plot. Source data are the same as for Supplementary Fig. 22A and B and are provided as a Source Data file.

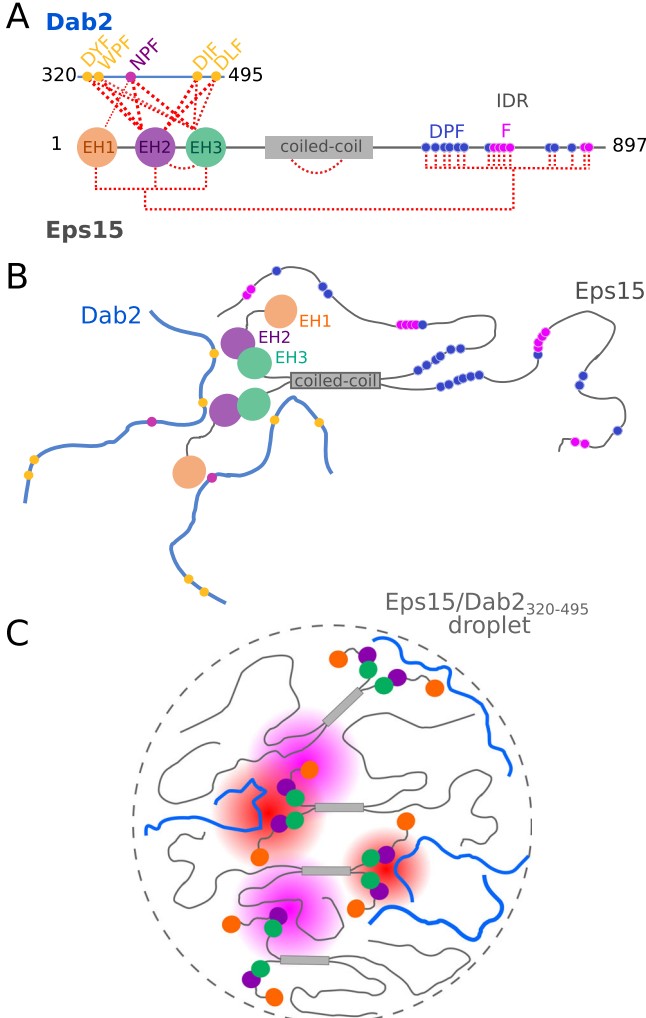

**Fig. 7 | Schematic of Eps15 and Dab2$_{320-495}$ interactions and phase separation. A** illustration of the complex interaction network between Eps15 and Dab2$_{320-495}$, including intra-molecular interactions within Eps15. Interactions are visualized with red dashed lines. **B** Cartoon of self-interactions within Eps15 and of interactions between Eps15 and Dab2$_{320-495}$. Eps15's EH domains interact with Eps15$_{IDR}$ and with Dab2$_{320-495}$. EH2 and EH3 tumble together. All three EH domains interact with the NPF motif within Dab2$_{320-495}$, while EH2 and EH3 are promiscuous binders. **C** The weak multivalent interaction network of the two IDRs with EH123 likely contributes to co-phase separation of Eps15 and Dab2$_{320-495}$. Eps15:Dab2 interactions are highlighted in red shading, and Eps15:Eps15 interactions are highlighted in pink shading.

The present promiscuity of interactions, particularly in the context of the full EH123 construct, suggests that the EH domains might bind to Eps15's own IDR, which has also been suggested previously[13,16,42]. Addition of Eps15$_{IDR}$ to $^{15}$N EH123 indeed led to CSPs and increased $^{15}$N R$_{1\rho}$ relaxation rates confirming this intra-molecular interaction (Fig. 5A). Our data recorded on $^{15}$N labeled Eps15$_{IDR}$ show that EH123 binds to all phenylalanine containing regions of Eps15$_{IDR}$,

highlighting the promiscuity of EH123 also observed in the binding to Dab2$_{320-495}$ (Fig. 5B). While one would expect that an NPF containing IDR, such as Dab2$_{320-495}$, might replace Eps15$_{IDR}$ from EH123 due to the preferred interaction of EH domains with NPF motifs compared to more promiscuous binding sites, this seems not to be the case. When both Eps15$_{IDR}$ and Dab2$_{320-495}$ are added to $^{15}$N labeled EH123, the spectrum and corresponding relaxation rates show signatures of binding to both proteins, although the individual contributions cannot be disentangled due to their similar interaction sites on EH123 (Fig. 5A). A similar behavior is observed when a sample containing all three proteins is detected from the side of $^{15}$N Eps15$_{IDR}$ or $^{15}$N Dab2$_{320-495}$. Only a few regions within the spectra of Eps15$_{IDR}$ and Dab2$_{320-495}$ slightly revert towards their unbound state when both EH123 and the respective other IDR (Dab2$_{320-495}$ or Eps15$_{IDR}$) are added (Fig. 5D, E), suggesting that both Eps15$_{IDR}$ and Dab2$_{320-495}$ can bind EH123 at the same time.

How is this possible when Dab2$_{320-495}$ and Eps15$_{IDR}$ occupy the same binding sites on EH123? Key to this question is likely the multivalency by which both IDRs interact with EH123, such that, while some motifs of Dab2$_{320-495}$ might be displaced from EH123 in a dynamic fashion when Eps15$_{IDR}$ is added—or vice versa—others maintain, thereby creating a dynamic trimeric (or even higher order) complex. Which of the motifs remain bound to EH123 and which get displaced is certainly a function of the individual motif's affinities. This is easiest visualized as a dynamic equilibrium where motifs in Dab2 are competing with Eps15's own IDR for binding to the EH domains, forming a complex and dynamic network (Fig. 7A, B). The fact that neither Dab2$_{320-495}$ nor Eps15$_{IDR}$ can effectively out-compete the other suggests that the strengths of the individual interactions between EH123 and Dab2$_{320-495}$ or Eps15$_{IDR}$ are not vastly different. The affinity between the EH domains and Eps15$_{IDR}$ might be increased in native full-length Eps15, as a result of proximity effects. However, our phase separation experiments with full-length Eps15 and Dab2$_{320-495}$ show recruitment of Dab2 into the Eps15 condensates, arguing for a physiological relevance of the interactions seen between Dab2$_{320-495}$, the EH domains and Eps15$_{IDR}$ and the competition effects observed.

While binding of EH domains to an intrinsically disordered linker of the same protein has been observed for the EH domain contained in EHD2[43], the interaction network established by Eps15 is certainly remarkable, since so many interactions take place and compete with each other (Fig. 7). This has implications for all other binding partners of Eps15, whether binding to the EH domains or Eps15's IDR, since any binding partner will be competing with these intra-molecular interactions. Indeed, the interactions observed between Eps15 EH domains and Eps15$_{IDR}$ could constitute some kind of auto-inhibitory mechanism, such as recently described for WW domain proteins[44], which could be (partially) released by other interaction partners, such as Dab2. In this context, it should be noted that the interactions between EH123 and Eps15$_{IDR}$ could be both intra-molecular within one Eps15 molecule and inter-molecular between different Eps15 molecules. Indeed, Eps15-Eps15 interactions have previously been suggested to drive Eps15 phase separation in the context of

clathrin-mediated endocytosis. For example, based on previous literature on the interaction between the EH domain of POB1 and DPF motifs[42], which are also contained in Eps15$_{IDR}$, interaction between Eps15 EH domains and Eps15$_{IDR}$ has been suggested to be important for liquid-liquid phase separation[16]. While our molecular data show that not only DPF motifs, but also other F-rich protein regions are involved in this interaction, other interactions of similar strengths between Eps15 and its binding partners may drive this partner into the liquid-like droplets, such as it has been observed for FCHo1/2[16] or ubiquitin[45]. Of note, phenylalanines have been identified as critical 'stickers' to promote weak interactions in liquid-liquid phase separation of IDPs[46]. Usually, they are thought to interact with phenylalanines in other IDPs, thereby creating a dynamic interaction network. In the case of Eps15, we have observed phenylalanines to contact small folded domains (EH domains)−interactions that may also drive Dab2$_{320-495}$ into Eps15 droplets. In good agreement with this hypothesis, we observe that Dab2$_{320-495}$, which does not form droplets on its own, also enters liquid-like droplets of Eps15, likely due to the weak multivalent network between the two (Fig. 7C). Therefore, while current literature points to Eps15 as the main initiator of condensate formation during the early phases of CME[16] our data suggest that Dab2 may be recruited into such condensates also in the cellular context. Interestingly, even though Eps15's IDR and Dab2$_{320-495}$ seem to compete for the same interaction sites on EH123, the presence of Dab2$_{320-495}$ had only negligible effects on the Eps15 condensates (Fig, 6 and Supplementary Figs. 22, 23) and even a large excess of Dab2$_{320-495}$ did not dissolve Eps15 droplets (Supplementary Fig. 23). This behavior may make sense in the context of the early clathrin coated pit: Eps15 condensates are indeed thought to function as an initiator[16], responsible to accumulate downstream client proteins that make endocytosis progress. Eps15 only moves away from the pit later in the process, when many other CLASPs/accessory proteins will have enriched at the pit[47,48]. It is thus possible that a much higher concentration of competitive interaction partners from different regions of Dab2, but also other NPF (and non-NPF) interaction partners are needed to affect phase separation or segregate these proteins out of Eps15 condensates. Indeed, it will be interesting to see whether the interactions between EH domains and Eps15$_{IDR}$ are maintained also in the presence of proteins containing multiple NPF motifs, binding with higher affinities[18], and what consequences this has on liquid-liquid phase separation at the endocytic pit and thus progression of productive endocytosis.

## Methods

### Cloning

Eps15-pmCherryN1 was a gift from Christien Merrifield (Addgene plasmid # 27696; http://n2t.net/addgene:27696; RRID: Addgene_27696)[48]. The three individual EH domains (EH1, EH2, EH3) and Eps15$_{IDR\ 761-896}$ were cloned into the pET41c vector leading to constructs with a non-cleavable C-terminal His-tag. Full length Eps15, EH123, Eps15$_{IDR}$, Eps15$_{IDR\ 481-581}$, Eps15$_{IDR\ 569-671}$ and Eps15$_{IDR\ 648-780}$ were cloned into the pET28a vector with an N-terminal GB1 solubility tag pET28-6His-GB1. This leads to the expression of a TEV (tobacco etch virus) cleavable 6His-GB1-TEV site construct followed by the protein of interest. The gene of Dab2$_{320-495}$ was purchased from Twist Bioscience and cloned into a pET28a vector with non-cleavable C-terminal His-tag. For the purpose of fluorescence labeling, a single cysteine mutant of Dab2$_{320-495}$ was constructed using site-directed mutagenesis (Dab2$_{320-495\ S328C}$). Dab2$_{328-360}$ and Dab2$_{358-390}$ were cloned into a pET28a vector with an N-terminal GB1 solubility tag pET28-6His-GB1. The UniProt IDs of Eps15 and Dab2 used in this study are P42566 and P98082, respectively. The primer list for all cloning performed can be found below:

| Restriction enzyme cloning | |
| --- | --- |
| Eps15_120_Xho1_rv | GTG GTG CTC GAG CTC AGC TGC AGA GGT TCC |
| Eps15_215_Xho1_rv | GTG GTG CTC GAG TCT CTT AGA TGG TGG CAC CAA GGC |
| Eps15_310_Xho1_rv | GTG GTG CTC GAG TGG TGG AAT CAT TTC AGG AGT AAG AAC GTG |
| Eps15_671_Xho1_rv | AGT GCC TCG AGT TAA GTG CTT GAA GTG GCA AAA GGA TCA G |
| Eps15_780_Not1_rv | GTG GTG GCG GCC GCT CTT GTT GGA GTT CCG ATC TTT GGT GG |
| Eps15_896_Not1_rv | GTG GTG GCG GCC GCC CGT GCT TCT GAT ATC TCA GAT TTG CTG AG |
| Dab2_360_Not1_rv | GCA CTG CGG CCG CTT ATT GGG CCT CTT GCT TCC CG |
| Dab2_390_Not_rv | GCA CTG CGG CCG CTT AGG AAA ATC CGT TTT GCT CGC GC |
| Eps15_1_Nde1_fw | GAT ATA CAT ATG GCT GCG GCG GC |
| Eps15_121_Nde1_fw | GAT ATA CAT ATG CCA TGG GCT GTA AAA CCT GAA GAT AAG GC |
| Eps15_216_Nde1_fw | GAT ATA CAT ATG AAA ACG TGG GTT GTA TCC CCT GCA G |
| Eps15_481_Nde1_fw | GAT ATA CAT ATG CAC CTA CAA GAT TCA CAA CAG GAA ATT AGT TCA ATG C |
| Eps15_761_Nde1_fw | GATATA CATATG TCG GTC AAA AGT GAA GAT GAA CCC CC |
| Eps15_569_Nde1_fw | GTG CGC ATA TGT CTG GTG TGA CTG ATG AAA ATG AGG TG |
| Dab2_328_Nde_fw | GTG CGC ATA TGA GCA CCC CCT GT CCA ATG G |
| Dab2_358_Nde_fw | GTG CGC ATA TGG AGG CCC AAG CGG GC |
| Q5® Site-Directed Mutagenesis Kit (NEB) | |
| Eps15-IDR1(481-581)_F | CTC GAG CAC CAC CAC CAC CAC C |
| Eps15-IDR1(481-581)_R | AAC AGC TGT AGT CAC CTC |
| Eps15-IDR1(648-780)_F | AAA GGT TCA GAT CCA TTT G |
| Eps15-IDR1(648-780)_R | CAT ATG ACC CTG GAA GTA C |
| Site directed mutagenesis | |
| Dab2_S328C_fw | CTA GCA GCT GTA CCC CCT TGT CCA ATG GTC C |
| Dab2_S328C_rv | CAA GGG GGT ACA GCT GCT AGA ACT ATT CTC CTT CTT CAT GGT ATA TC |

### Protein expression and purification

The proteins were expressed in the *E. coli* Rosetta (DE3) strain and grown in LB medium with 30 mg/L Kanamycin and 30 mg/L Chloramphenicol at 37 °C. When the optical density (OD) at 600 nm was around 0.6−1, expression was induced with 1 mM isopropyl-β-D-thiogalactopyranoside (IPTG) and the expression was then continued at 20 °C overnight. For isotope labeling ($^{15}$N, $^{13}$C) M9 minimal medium was used and supplemented with 1 g/L $^{15}$NH$_4$Cl and/or with 2 g/L $^{13}$C-glucose.

Cells were lysed by sonication in lysis buffer (20 mM Tris, 150 mM NaCl pH 8, with Roche Ethylenediaminetetraacetic Acid (EDTA)-free protease inhibitor cocktail (Sigma-Aldrich Chemie GmbH)). Purification involved a two-step process: initial nickel purification followed by size-exclusion chromatography (SEC). The nickel column was equilibrated in lysis buffer before the filtered lysate was applied to a nickel column. The column was then washed with lysis buffer containing 20 mM imidazole, and the protein was eluted using lysis buffer supplemented with 400 mM imidazole. For all constructs expressed in the pET28-6His-GB1 vector, the eluted fraction, which contained the target protein (validated by SDS-PAGE and Coomassie staining), underwent

overnight dialysis with 1 mg TEV protease at 4 °C in 500 mL of lysis buffer supplemented with 5 mM β-mercaptoethanol before proceeding to the SEC purification step. Proteins were further purified using a Superdex 75 or a Superdex 200 column, equilibrated in NMR buffer (50 mM Na-phosphate pH 6, 150 mM NaCl, and 2 mM dithiothreitol (DTT)). Fractions containing pure protein (validated by SDS-PAGE) were concentrated and frozen with final protein concentrations determined by absorbance at 280 nm using extinction coefficients determined by Expasy Protparam[49].

## NMR spectroscopy

NMR experiments were measured at the Leibniz-Forschungsinstitut für Molekulare Pharmakologie (FMP), Berlin, Germany ($^1$H frequencies of 600, 750, 900, 1200 MHz), and at the Institute of Structural Biology (IBS), Grenoble, France ($^1$H frequencies of 600 MHz). The spectrometers were equipped with either room-temperature- (750 MHz) or cryo-probes (600, 900, 1200 MHz). All experiments were measured in NMR buffer (50 mM Na-phosphate pH 6, 150 mM NaCl, 2 mM DTT) at 25 °C. Addition of up to 5 mM CaCl$_2$ did not affect our results. Spectra were processed with NMRPipe[50], using qMDD[51] for non-uniformly sampled assignment spectra, and analyzed with CCPN[52]. $^1$H-$^{15}$N HSQC, TROSY-HSQC and triple resonance experiments were acquired using NMRlib[53]. The software TopSpin 3.5 and 4.4.1 (Bruker) were used for data acquisition.

## $^{15}$N, $^{13}$C Backbone assignment

Almost complete assignments could be obtained for all protein constructs. The assignment spectra of the $^{15}$N, $^{13}$C labeled EH1, EH2, EH3, Dab2$_{328-360}$, and Dab2$_{358-390}$ were acquired at a $^1$H frequency of 750 MHz. Assignment spectra of Eps15$_{IDR}$, Eps15$_{IDR\ 480-581}$, Eps15$_{IDR\ 569-671}$, Eps15$_{IDR\ 648-780}$, Eps15$_{IDR\ 761-896}$, and Dab2$_{320-495}$ were acquired at a $^1$H frequency of 600 MHz. Standard BEST-TROSY triple resonance experiments correlating CO, Cα, Cβ resonances (HNCO, HNCOCA, HNCA, iHNCA, HNCOCACB, iHNCACB)[54] were acquired. The assignments were done in CCPN[52] and then validated using MARS[55]. Secondary chemical shifts and secondary structure propensities[29] were calculated using random coil values from refDB[56]. Overlapping the 3 TROSY-HSQC spectra of the individual EH domains (EH1, EH2, EH3) we transferred the $^1$H and $^{15}$N resonances to the TROSY-HSQC of the EH123 domain. The assignment of the different Eps15$_{IDR}$ stretches was transferred to the spectrum of the full Eps15$_{IDR}$ in a similar way. Dab2$_{320-495}$ comprises many prolines, and some peaks within its $^1$H-$^{15}$N HSQC spectrum could be attributed to appear due to cis/trans isomerization of neighboring prolines. We have not observed differences in binding due to cis/trans proline isomerization for the interactions tested and therefore all plots within the manuscript refer to the main conformational state of the protein.

## Titrations, $^{15}$N relaxation, and relaxation dispersion

Extraction of peak intensities (I) as well as $^1$H and $^{15}$N chemical shifts, were carried out from $^1$H-$^{15}$N HSQC or BEST-TROSY-HSQC spectra. Combined chemical shift perturbations (CSPs) were calculated using

$$CSP = \sqrt{\left(\delta^1H \cdot 6.5\right)^2 + \left(\delta^{15}N\right)^2} \tag{1}$$

The specific concentrations used in the different titrations are indicated in the respective figures, with the percentage of partner indicating the molar ratio of partner protein:observed protein.

Residue specific $K_d$ values were estimated by fitting the following equation to the CSPs as a function of concentration:

$$\Delta CSP = \Delta CSP_{max} \cdot \frac{[Dab2] + [EHx] + Kd - \sqrt{\left([Dab2] + [EHx] + Kd\right)^2 - 4 \cdot [Dab2] \cdot [EHx]}}{2 \cdot [Dab2]} \tag{2}$$

Fitting was performed with Python.

$^{15}$N R$_{1\rho}$, R$_1$ and {$^1$H}-$^{15}$N HetNOE relaxation rates[57] were assessed at 600 MHz and 1200 MHz $^1$H Larmor frequency. The spin-lock field for the R$_{1\rho}$ experiment was set to 1500 Hz or 2000 Hz for the 600 MHz and 1200 MHz magnets, respectively, and 6–7 delays, between 10 and 230 ms (for the disordered Dab2$_{320-495}$ and Eps15$_{IDR}$) and between 10 and 70 ms (for the folded EH domains) were used to sample the decay of magnetization. The relaxation rates were determined from an exponential fit to the peak intensity relative to the delay time. Errors of the fitted rates were derived from the experimental uncertainty. Inter scan delays were typically of 1.9 s. R$_1$ was measured using 6–7 delays between 0 to 1.2 s with an inter-scan delay of typically 1.9 or 1.3 s. Inter scan delays of HetNOE experiments were 1 s.

Relaxation dispersion experiments[58] were conducted at 600 and 900 MHz, employing 14 CPMG frequencies ranging from 31 to 1000 Hz, with a constant-time relaxation of 32 ms. R$_2$ uncertainty in the CPMG experiments was estimated via Monte Carlo sampling from a normal distribution based on the experimental noise. The error bars represent the 15.9th and 84.1st percentiles of the resulting R$_2$ distribution and were determined by the software ChemEx (https://github.com/gbouvignies/chemex). For the experiment with 100 μM Dab2$_{320-495}$ in the presence of 10% EH2, 12 residues within the helix$_N$ and the NPF motif (Val340, Asp341, Gln345, Gln346, Gln349, Ser351, Thr354, Lys356, Phe366, Phe 398, Val399, Asp426) were fit globally using a 2 site exchange model with the software ChemEx (https://github.com/gbouvignies/chemex). An exchange rate of 149 ± 13 s$^{-1}$ and a percentage bound of 3 ± 0.3% were obtained from the fit.

## Conformational ensemble of Dab2$_{320-495}$

A conformational ensemble was calculated from Dab2$_{320-495}$ and Dab2$_{328-360}$ using a combination of the statistical coil generator, flexible meccano[24], and the genetic algorithm ASTEROIDS[25]. From a statistical coil ensemble of 10,000 conformers, 200 conformations that together best described the H$_N$, N, CO, Cα and Cβ chemical shifts of the proteins were selected. A new ensemble of 8500 conformers was generated using the Φ and ψ angles from the previous selection and supplemented with 1500 conformations from the previous pool of conformers. A new selection of 200 conformations was performed on the new pool. This iteration was repeated 4 times. Ensemble-averaged chemical shifts were generated using SPARTA[59], and secondary chemical shifts were calculated based on RefDB[56].

## Structures of the individual EH domains

The structures that were used to plot the CSPs for EH2 and EH3 are the PDB codes 1FF1[31] and 1C07[32], respectively. For EH1, we use an AlphaFold2[34,60] prediction since at the time of writing no *human* Eps15 EH1 structure was deposited in the PDB. The CSPs of each EH domain with 100 μM Dab2$_{320-495}$ (100%) were plotted onto the structures. The linear gradient is from 0%, which represents no CSP, up to 100%, which is the highest CSP observed at 0.75 ppm. The structures were visualized using Pymol.

## Protein labeling with fluorescent dyes

Protein labeling was performed using maleimide dyes; AZDye488 (Vectorlabs, CA, USA), for Dab2$_{320-495\ S328C}$ and AZDye594 (Vectorlabs, CA, USA) for Eps15 at random cysteine side chains, essentially as described previously[61,62]. After adding 10 mM DTT overnight to fully reduce the proteins, they were dialyzed into a phosphate-buffer (50 mM Na-phosphate pH 7, 150 mM NaCl) for 2 × 1 h at room temperature. At least 5 times molar excess of the dyes was used for the labeling reaction, which, after mixing, contained a maximum of 10% V/V DMSO. The protein/dye mixtures were allowed to react for 1 h at room temperature and incubated overnight at 4 °C. To stop the reaction, 10 mM of DTT were added before injecting the mixtures on a SEC70 or 650 equilibrated in NMR buffer to remove any unconjugated dye.

**Article**

## Protein droplets and microscopy imaging

All droplet formation assays were performed in a buffer of 50 mM Na-phosphate pH 6, 150 mM NaCl, and 2 mM DTT, 3% w/v PEG8000, using different ratios of Eps15 (0.5 μM to 10 μM) and $Dab2_{320-495}$ (0 μM to 140 μM). For both proteins, a ratio of 10:1 of non-labeled: fluorescently labeled protein was used. A total sample volume of 20 μL was placed in 8-well polystyrene chambers with 1.5 borosilicate coverglass (Nunc Lab-Tek). Epi-fluorescence imaging was performed with a Nikon Ti Eclipse microscope with a 60x (PLAN APO, NA:1.40, WD 0.13 mm) oil immersion objective. The setup was controlled by the imaging software NIS (Nikon). Confocal microscopy was performed with the LSM780, and the setup was controlled by Zeiss Zen Black software. $Dab2_{320-495\ S328C}$ AZDye488 (Ex.: 488 nm; EmF.: 490–535 nm) and Eps15 AZDye594 (Ex.: 561; EmF.: 569–631 nm) were imaged. A PL APO DIC M27 63×/1.40 NA oil objective (Carl Zeiss Microscopy) at a zoom factor of 1 or 3 and a line average of 1 was used, acquiring images of 512 × 512/1024 × 1024 pixels, respectively. Microscopy images were imported and analyzed using ImageJ/Fiji[63]. The intensity ratio of the droplets versus background was calculated by manually selecting droplets and comparing the intensity to an identical area of the background. The final ratio was calculated as an average over all droplets picked across the different images for each condition. The number and size of the droplets were determined by applying an Otsu threshold to each image. The number of droplets was counted in each individual image for each condition, while the size of the droplets was calculated as an average of droplet size across all images of each condition. For the determination of the droplet size and count as well as the partitioning of Eps15 and $Dab2_{320-495}$ into droplets, images at a zoom factor of 1 (512 × 512 pixels) were used, and the analyses from ImageJ/Fiji were plotted and visualized using Python. The number of images included in the analysis for each condition can be found in Fig. 6. The images were acquired at different spatial positions of the same samples, respectively.

## Reporting summary

Further information on research design is available in the Nature Portfolio Reporting Summary linked to this article.

## Data availability

All study data are included in the article, supporting information, the Source Data files and/or can be obtained from the corresponding author upon request. The chemical shift assignments generated in this study have been deposited in the Biological Magnetic Resonance Data Bank (BMRB) under the accession numbers 52613 ($Dab2_{320-495}$), 52866 ($Eps15_{IDR\ 481-581}$), 52864 ($Eps15_{IDR\ 569-671}$), 52863 ($Eps15_{IDR\ 648-780}$), and 52867 ($Eps15_{IDR\ 761-896}$), respectively. The NMR and imaging statistics generated in this study are provided in the Source Data files. The PDB entries of EH2 and EH3 used in this work are: 1FF1 (EH2) and 1C07 (EH3). Source data are provided with this paper.

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

## Acknowledgments

We thank all members of the Milles group for fruitful discussions and critical proofreading. We thank M.R. Jensen for providing the pET28-6His-GB1 plasmid, M. Blackledge and M.R. Jensen for providing NMR analysis scripts, P. Schmieder, N. Trieloff and M. Beerbaum for technical assistance on the NMR spectrometers and T. Soykan, M. Biek and M. Lehmann from the Imaging Facility of FMP for technical support and assistance with image analysis. The Institut de Biologie Structurale acknowledges integration into the Interdisciplinary Research Institute of Grenoble. This work was supported by the Leibniz-Forschungsinstitut für Molekulare Pharmakologie (FMP) (to S.M.). This project has received funding from the European Research Council (ERC) Starting Grant MultiMotif to S.M. under the European Union's Horizon 2020 research and innovation program (grant agreement no. 802209).

## Author contributions

S.M., A.P., and I.M.V. designed the research. A.P., I.M.V., K.M., M.T., and A.S. performed research. A.P., I.M.V., and S.M. analyzed the data. A.P., I.M.V., and S.M. wrote the paper.

## Funding

## Competing interests

The authors declare no competing interests.
