## [Transparent Peer Review file · Nature Communications]

Promiscuous and multivalent interactions between Eps15 and partner protein Dab2 generate a complex interaction network.

Corresponding Author: Dr Sigrid Milles

Version 0:

Reviewer comments:

Reviewer #1

(Remarks to the Author)

Review of Nature Communications Manuscript NCOMMS-25-15030

Title: Promiscuous and multivalent interactions between Eps15 and partner protein Dab2 generate a complex interaction network.

First Authors: Papagiannoula and Vedel

Summary of key results: This paper by Papagiannoula and Vedel et al. used NMR spectroscopy to characterize the interactions between the endocytic protein Eps15 and one of its binding partners, Dab2. The authors find that while the three EH domains of Eps15 display the expected interaction with the canonical NPF motif in the intrinsically disordered region (IDR) of Dab2, these EH domains also bind many other phenylalanine-containing motifs throughout the Dab2 IDR. The first EH domain is the most selective in binding the NPF motif, while the remaining two EH domains are more promiscuous in their binding behavior. This binding promiscuity enables the Eps15 EH domains to interact with Eps15's own IDR despite a lack of NPF motifs, establishing a complex interaction network between Eps15 and Dab2 in which the EH domains can simultaneously bind both IDRs. The authors show that this interaction network may enable Dab2 to partition into Eps15 condensates. The NMR data are particularly impressive and convincing, and the paper is well-written and presented. The condensate data at the end of the paper would benefit from some additional experiments and quantification, described below in more detail. Overall, this reviewer feels that the paper offers an important contribution to our understanding of the multicomponent protein interaction networks that drive clathrin-mediated endocytosis, and recommends publication after addressing the comments below.

Main points:

Point 1. The direct interaction observed between Dab2 and Eps15 suggests that inclusion of Dab2 should alter Eps15 condensation and change the Eps15 saturation concentration (C_{sat}). However, it is unclear whether Dab2 should strengthen or weaken Eps15 condensation, as the two proteins can directly interact but this interaction competes with Eps15 self-interactions. Can the authors provide a more extensive characterization of Eps15 phase behavior at varying Eps15 concentrations in the presence or absence of Dab2, in order to determine whether Dab2 alters Eps15 C_{sat}? After characterizing the impacts of Dab2 on Eps15 C_{sat}, can the authors provide readers with some interpretation of how Dab2 alters Eps15 condensates in the context of early CCPs?

Point 2. Passive client molecules can sometimes partition and enrich within condensates without directly interacting with the scaffold protein of the condensate. Can the authors provide some controls using molecules not expected to interact with Eps15 (e.g. GFP, labeled Dextran) to further support Dab2's partitioning as arising from a direct interaction with Eps15?

Point 3. Some quantification of the condensate microscopy data would help in interpreting and understanding the partitioning behavior of Dab2. Can the authors quantify condensate size and dense phase Eps15 and Dab2 intensities and plot those parameters as a function of Eps15:Dab2 stoichiometry?

Point 4. How did the authors choose the Eps15:Dab2 stoichiometries of 1:1 and 1:20? Do either of these stoichiometries reflect their physiological stoichiometry in the cell?

Minor points:

Minor point 1. Panel 2D shows profiles of Y342, but the text only discusses V399. Is the statement “The other interacting regions show very little broadening and are mainly characterized by CSPs” in reference to the Y342 data in panel 2D? If so, can the authors be more explicit about the specific residue they are referring to?

Minor point 2. Consider re-ordering panels 2C, D, and E to reflect their order in the text.

Minor point 3. The text references to the different panels in Fig. 3 are confusing. There are no explicit references to panels 3A-C or 3E, but two references simply to “Fig. 3”. Please update the text with the specific references to each panel in the order in which they appear in the figure.

Minor point 4. The R1ρ plot in Fig. 5A shows EH1 data in the presence of Eps15IDR and Eps15IDR + Dab2320-495, but no data for Dab2320-495 alone. However, the CSP plot shows data for all three conditions. Why is there no EH1 R1ρ data for Dab2320-495 alone?

Reviewer #2

(Remarks to the Author)
Please see file attached.

[Editorial Note: The attachment has been appended to the end of the file]

Reviewer #3

(Remarks to the Author)

This exciting manuscript from Papagiannoula and colleagues provides a detailed glimpse into the complexity of the dynamic, multivalent interaction landscape of two intrinsically disordered proteins involved in clathrin-mediated endocytosis, Eps15 and Dab2. Using NMR spectroscopy, which is exceptionally well-suited for characterizing these highly dynamic disordered proteins, the authors demonstrate the importance of phenylalanine residues both within and outside of previously identified binding motifs in mediating intermolecular (Eps15-Dab2 and Eps15-Eps15) interactions as well as facilitating intramolecular interactions between the Eps15 IDR and the EH domains. The data are presented clearly and logically and thoroughly support the authors' conclusions. This work expands current knowledge of protein interactions in the early stages of endocytosis and will be of great interest to many.

I have only a few minor questions and comments that I hope can be addressed to provide additional clarity on the important concepts and data presented in the manuscript:

1. The experiments comparing the interactions of Dab2 with the isolated EH domains and the EH123 construct clearly demonstrate that there is benefit to having 3 of these domains linked, presumably due to avidity effects that enhance the binding affinity. In the competition experiments, the data show that the Eps15 IDR can partially outcompete Dab2 for binding to the EH domains when added in trans...but might this competition be more effective in the context of full-length Eps15? It is easy to envision that interactions of the Eps15 IDR with the EH domains would be even more favorable in the context of the full-length protein. How do the authors think this would impact binding to Dab2 and other interaction partners? Some further discussion of how the intramolecular interactions of Eps15 may differ in the intact protein versus in mixtures of truncated forms is warranted.

2. I am curious if the authors have attempted to determine the binding affinities by any method other than NMR. Might there be tighter interactions that aren't quantifiable by NMR (ie. for resonances that are broadened beyond detection)?

3. The text labels in many of the figures are far too tiny to be legible, even at high magnification. This is particularly true for Figure 4C, Figure S6, and Figure S7, but larger labels would generally be helpful throughout.

Version 1:

Reviewer comments:

Reviewer #1

(Remarks to the Author)

The authors have done a commendable job in responding to my review. I feel that the condensate portion of the manuscript has been particularly strengthened by the addition of their new data. I have no further comments or questions. Thank you to the authors for their careful response to my points, and I look forward to seeing the paper published in Nature Communications.

Reviewer #3

(Remarks to the Author)

The authors have adequately addressed all of my previous concerns. The additions they have made to the manuscript in response to the other reviewers' comments greatly enhance the manuscript and improve the accessibility of the findings to a broader audience. I remain enthusiastic about this work and have no further comments or suggestions.

Response to the reviewers

Promiscuous and multivalent interactions between Eps15 and partner protein Dab2 generate a complex interaction network.

Authors

Papagiannoula[#], Vedel^{1#}, *et al.*

[#]equal contribution

[*milles@fmp-berlin.de](mailto:milles@fmp-berlin.de)

We appreciate the reviewers' thoughtful assessment of our manuscript and their valuable feedback. We have carefully considered all comments and suggestions, which have helped us to substantially improve the manuscript. Our detailed responses to each point are provided in blue font, and all corresponding changes in the manuscript are highlighted in yellow.

Reviewer #1 (Remarks to the Author):

Review of Nature Communications Manuscript NCOMMS-25-15030

Title: Promiscuous and multivalent interactions between Eps15 and partner protein Dab2 generate a complex interaction network.

First Authors: Papagiannoula and Vedel

Summary of key results: This paper by Papagiannoula and Vedel et al. used NMR spectroscopy to characterize the interactions between the endocytic protein Eps15 and one of its binding partners, Dab2. The authors find that while the three EH domains of Eps15 display the expected interaction with the canonical NPF motif in the intrinsically disordered region (IDR) of Dab2, these EH domains also bind many other phenylalanine-containing motifs throughout the Dab2 IDR. The first EH domain is the most selective in binding the NPF motif, while the remaining two EH domains are more promiscuous in their binding behavior. This binding promiscuity enables the Eps15 EH domains to interact with Eps15's own IDR despite a lack of NPF motifs, establishing a complex interaction network between Eps15 and Dab2 in which the EH domains can simultaneously bind both IDRs. The authors show that this interaction network may enable Dab2 to partition into Eps15 condensates. The NMR data are particularly impressive and convincing, and the paper is well-written and presented. The condensate data at the end of the paper would benefit from some additional experiments and quantification, described below in more detail. Overall, this reviewer feels that the paper offers an important contribution to our understanding of the multicomponent protein interaction networks that drive clathrin-mediated endocytosis, and recommends publication after addressing the comments below.

We thank the reviewer for the very positive assessment of our work and we provide detailed replies to the different individual points below.

Main points:

Point 1. The direct interaction observed between Dab2 and Eps15 suggests that inclusion of Dab2 should alter Eps15 condensation and change the Eps15 saturation concentration (C_{sat}). However, it is unclear whether Dab2 should strengthen or weaken Eps15 condensation, as the two proteins

can directly interact but this interaction competes with Eps15 self-interactions. Can the authors provide a more extensive characterization of Eps15 phase behavior at varying Eps15 concentrations in the presence or absence of Dab2, in order to determine whether Dab2 alters Eps15 Csat? After characterizing the impacts of Dab2 on Eps15 Csat, can the authors provide readers with some interpretation of how Dab2 alters Eps15 condensates in the context of early CCPs?

The reviewer raises an important point. We have shown that the EH domains within Eps15 interact with the IDR of Dab2 through its NPF motifs, but also through other hydrophobic residues, in a promiscuous fashion. This promiscuity allows Eps15's EH domains not only to bind client proteins containing NPF motifs, such as Dab2, but also its own IDR, which is devoid of NPF motifs. The same binding regions on the EH domains are involved in Dab2 binding and Eps15 IDR binding, suggesting that the interaction should be competitive – a question we addressed in Figure 5 using NMR spectroscopy. Surprisingly, we see only little competition (Fig. 5D and E), likely because the binding mode is very dynamic and association transient, such that EH domains associate and dissociate from both Eps15 IDR and Dab2 very rapidly, allowing an overall binding to both IDRs at the same time.

In a condensate, formed by Eps15 through the interactions between its EH domains and its IDR, the presence of a competitive partner is expected to influence at which concentrations condensates can still be formed and when this is no longer the case. We have previously addressed this question by adding a large excess of Dab2 (20-fold) to Eps15 condensates (former Fig. 6D, now Supplementary Fig. 23), which, however, did not lead to dissolution of the Eps15 condensates.

Motivated by the reviewer's insightful suggestion, we now addressed different concentrations of Eps15 with different concentrations of Dab2 as a client protein for droplet formation and also assessed the lower end of the concentration range that would lead to droplet formation (PbyP Fig. 1 and new Fig. 6 in the revised manuscript). For this, we varied Eps15 concentrations from 0.5 μM to 10 μM and assessed droplet formation in the absence and presence of 7 μM Dab2₃₂₀₋₄₉₅. In addition, at a concentration of 7 μM Eps15, we added Dab2₃₂₀₋₄₉₅ at concentrations from 0.5 μM to 10 μM .

Overall, the number of droplets in the presence of Dab2₃₂₀₋₄₉₅ seems mildly lower, while the overall droplet size appears a little larger (PbyP Fig. 2, as well as Fig. 6 and Supplementary Fig. 22A, B of the revised manuscript). Given the large error bars resulting from an analysis over multiple images (see exact PbyP Fig. 2A, or in revised manuscript Fig. 6 for exact numbers), the effects are, however, marginal. Only at an Eps15 concentration of 0.5 μM we had difficulties finding any droplets in the presence of 7 μM Dab2. However, the condensates under these low Eps15 concentrations were already hard to find and very small in the absence of Dab2₃₂₀₋₄₉₅. Taken together, it thus seems like this construct of Dab2 (320-495) can be described as a client protein that competes with the interaction between EH domains and Eps15_{IDR} so minimally, that no competitive effect can be observed in the context of condensate formation for a wide range of protein concentrations.

PbyP Figure 1: Condensate formation of Eps15 in the presence and absence of Dab2₃₂₀₋₄₉₅. (A) Representative droplets imaged from Eps15 alone at different concentrations (0.5 μM to 10 μM , indicated above). (B) Eps15 droplets formed under the same conditions in the presence of 7 μM Dab2₃₂₀₋₄₉₅. (C) Eps15 droplets formed with 7 μM protein, to which different concentrations of Dab2₃₂₀₋₄₉₅ (0.5 μM to 10 μM , indicated above) was added. 10% of the protein used in these samples were fluorescently labeled. Eps15 was labeled at random cysteine side chains using AZDye 594-maleimide. Dab2₃₂₀₋₄₉₅ S328C was labeled at the introduced cysteine side chain (residue 328) using AZDye 488-maleimide. The scale bar is 5 μm .

In the context of early clathrin coated pits, this may be a wanted effect: In clathrin mediated endocytosis, Eps15 condensates acts as initiator,¹ responsible to accumulate downstream client proteins that make endocytosis progress. Eps15 only moves away from the pit later in the process, when many other CLASPs/accessory proteins will have enriched at the pit.^{2,3} It is thus possible that a much higher concentration of competitive interaction partners from different regions of Dab2, but also other NPF (and non-NPF) interaction partners are needed to affect phase separation or segregate these proteins out of Eps15 condensates. We indeed find this discussion very stimulating, thank the reviewer again for bringing this up, and included it also in the discussion of the revised manuscript (page 14).

While experimentally accessing the complex protein composition at the growing endocytic pit is beyond the scope of this article, it is nevertheless an exciting question—one that we are actively planning to pursue in future work.

PbyP Figure 2: The effect of protein concentration on droplet size and droplet count. (A) Droplet sizes plotted in a histogram for different concentrations of Eps15 and Dab2 as indicated above the plots. **(B)** The average droplet size plotted against Eps15 concentration in the presence and in the absence of 7 μM Dab2₃₂₀₋₄₉₅. The averages from a number of images are shown (see Supplementary Fig. 22 for exact numbers). **(C)** The droplet count/image is plotted against Eps15 concentration in the presence and absence of 7 μM Dab2₃₂₀₋₄₉₅. One point in the plot corresponds to the count in one image. Color codes as indicated in (B).

Point 2. Passive client molecules can sometimes partition and enrich within condensates without directly interacting with the scaffold protein of the condensate. Can the authors provide some controls using molecules not expected to interact with Eps15 (e.g. GFP, labeled Dextran) to further support Dab2's partitioning as arising from a direct interaction with Eps15?

We thank the reviewer for raising this important point, which is particularly relevant when weak interactions play a role for partitioning into the droplets. We have thus selected a molecule, which we expected not to interact with Eps15: a 38 base pair single stranded DNA molecule labeled with Atto488. We found that condensates formed by 7 μM Eps15, enriched with AZDye 594-labeled

protein, excluded the labeled DNA (PbyP Fig. 3 and Supplementary Fig. 23E), showing that the direct interaction between Dab2 and Eps15 is needed for Dab2 to enrich the condensates. Exclusion of the DNA is particularly remarkable since the molecule is actually comparatively small.

PbyP Figure 3: Exclusion of fluorescently labeled DNA from Eps15 condensates. Condensates were formed from 7 μM Eps15 supplemented with 10% fluorescently labeled Eps15 with AZDye 594-maleimide. Atto488-labeled 38 base pair DNA was then added to a concentration of 2 μM . Shown are images of the individual channels and the merged image. The scale bar is 5 μm .

Point 3. Some quantification of the condensate microscopy data would help in interpreting and understanding the partitioning behavior of Dab2. Can the authors quantify condensate size and dense phase Eps15 and Dab2 intensities and plot those parameters as a function of Eps15:Dab2 stoichiometry?

We thank the reviewer for proposing to look at the droplet behavior under different Eps15:Dab2 ratios. We have performed a large number of imaging experiments using droplets made from different Eps15 and Dab2₃₂₀₋₄₉₅ concentrations as described above. Droplet size (and number of droplets) seem to be mainly determined by the concentration of Eps15 (PbyP Fig. 1) and seem – if at all – only minimally modulated by the presence of Dab2₃₂₀₋₄₉₅. We think that this is a consequence of the relatively short construct of Dab2 used in our experiments (320-495), while in the cell the full length Dab2 and many other endocytic CLASPs will interact with Eps15, potentially altering Eps15's phase separation or co-phase separation under certain conditions.

We have then used the droplet images acquired at different Eps15 and Dab2₃₂₀₋₄₉₅ concentrations and assessed the partitioning of both Eps15 and Dab2₃₂₀₋₄₉₅ into the condensates by looking at the ratio of fluorescence intensity inside and outside of the droplets. We have plotted this ratio as a function of Eps15 concentration, Dab2₃₂₀₋₄₉₅ concentration, and against the concentration ratio between the two proteins (PbyP Fig. 4 and Supplementary Fig. 22). Overall, the relative fraction of protein inside versus outside the droplets increases with increasing protein concentration of the same kind. Eps15, for example, partitions more strongly into the droplets with increasing Eps15 concentration, and Dab2₃₂₀₋₄₉₅ partitions more strongly into droplets with increasing Dab2 concentrations. While the concentration of Dab2₃₂₀₋₄₉₅ does not seem to strongly affect Eps15 partitioning, Eps15 concentration does seem to positively affect partitioning of Dab2₃₂₀₋₄₉₅ into the droplets to a small extent. This probably reflects the increased reservoir of Eps15 condensates (in number or size) to be available for taking up Dab2₃₂₀₋₄₉₅ as a client protein.

PbyP Figure 4: Partitioning of Eps15 and Dab2₃₂₀₋₄₉₅ into droplets of different composition. (A) Fluorescence intensity ratios inside versus outside the droplets plotted against the Dab2₃₂₀₋₄₉₅ concentration. Shown are ratios in the green (Dab2) and the red (Eps15) channel. Eps15 was concentrated to 7 μ M through all experiments shown. **(B)** Fluorescence intensity ratios inside versus outside the droplets plotted against the Eps15 concentration. Shown are ratios in the green (Dab2) and the red (Eps15) channel at a Dab2₃₂₀₋₄₉₅ concentrated of 7 μ M and 0 μ M. **(C)** Fluorescence intensity ratios inside versus outside the droplets plotted against the concentration ratio of Dab2₃₂₀₋₄₉₅ versus Eps15 in the red and the green channel. The different points at a ratio of 0 stem from those droplets that were formed with different concentrations of Eps15.

We have included this analysis as new Supplementary Figs. 22C-E and discuss the related conclusions in the revised manuscript (pages 10/11).

Point 4. How did the authors choose the Eps15:Dab2 stoichiometries of 1:1 and 1:20? Do either of these stoichiometries reflect their physiological stoichiometry in the cell?

We had initially chosen the Eps15:Dab2 stoichiometry of 1:1, because this corresponded to the molar ratio between the proteins that we used in our NMR competition experiments. We then aimed to investigate whether higher concentrations of Dab2₃₂₀₋₄₉₅, harboring an NPF motif, can compete out the more promiscuous interactions between the Eps15 EH domains and Eps15's IDR and thus affect droplet formation. 1:20 provided a large excess of Dab2₃₂₀₋₄₉₅ and was, at the same time, biochemically still feasible. In this revised manuscript, however, we included a much more extensive study, analyzing the effect of different Eps15 and Dab2₃₂₀₋₄₉₅ concentrations on droplet formation with a thorough investigation of the saturating concentration of Eps15, condensate size and number as well as partitioning of both proteins, as suggested by the reviewer. We are grateful to the reviewer to propose these experiments, which we think strengthen our work a lot and provide more detailed insights into a possible working mechanisms on condensates in the context of endocytosis. The new data can be found in the revised Fig. 6, and Supplementary Fig.22. We maintained the data from the original Fig. 6 as Supplementary Fig. 23.

Minor points:

Minor point 1. Panel 2D shows profiles of Y342, but the text only discusses V399. Is the statement "The other interacting regions show very little broadening and are mainly characterized by CSPs" in reference to the Y342 data in panel 2D? If so, can the authors be more explicit about the specific residue they are referring to?

Former Fig. 2D and 2E (now changed to 2C and 2D) comprise examples for residues that do and do not show contributions towards intermediate exchange and corresponding line broadening. In the new version, we have now replaced Y342 by L486, since Y342 in fact does show a small contribution of exchange in the microsecond to millisecond range. To make our decisions to show these two residues clearer, we have inserted a corresponding explanation in the revised manuscript (page 4/5)

Minor point 2. Consider re-ordering panels 2C, D, and E to reflect their order in the text.

We thank the reviewer for this suggestion and have adapted the order of the figure panels in Fig. 2 accordingly. Former 2D and 2E are now 2C and 2D, former 2C has been named 2E.

Minor point 3. The text references to the different panels in Fig. 3 are confusing. There are no explicit references to panels 3A-C or 3E, but two references simply to "Fig. 3". Please update the text with the specific references to each panel in the order in which they appear in the figure.

We thank the reviewer for bringing this up. We have now incorporated specific citations to panels A-C and E of Fig. 3.

Minor point 4. The $R_{1\rho}$ plot in Fig. 5A shows EH1 data in the presence of Eps15IDR and Eps15IDR + Dab2320-495, but no data for Dab2320-495 alone. However, the CSP plot shows data for all three conditions. Why is there no EH1 $R_{1\rho}$ data for Dab2320-495 alone?

The plot showing $R_{1\rho}$ experiments in Fig. 5A shows data acquired on ^{15}N labeled EH123. In the presence of equal concentrations of Eps15_{IDR}, peaks corresponding to EH2 and EH3 are not visible any more in the spectrum of EH123, likely due to strongly slowed down tumbling times, which is why we can only show the rates of EH1 within the EH123 complex for samples that include Eps15_{IDR}. In the presence of both equal amounts of Dab2₃₂₀₋₄₉₅ and Eps15_{IDR}, the $R_{1\rho}$ rates of EH1 within the construct of EH123 are slightly higher than in the presence of only Eps15_{IDR}, showing that Dab2₃₂₀₋₄₉₅ does not fully compete off Eps15_{IDR} from binding to EH123 but rather manages to bind at the same time. We concluded this, because while Dab2₃₂₀₋₄₉₅ alone also leads to increased tumbling times of EH123, it does not broaden the peaks to disappearance (shown in Fig. 4C). Overlaying this $R_{1\rho}$ of EH123 in the presence of Dab2₃₂₀₋₄₉₅ is thus not necessary to draw the conclusion that both IDRs (Eps15_{IDR} and Dab2₃₂₀₋₄₉₅) can bind at the same time. However, for more completeness, we have now added all relaxation rates recorded at 1200 MHz in Supplementary Fig. 15B.

We thank the reviewer for pointing out that this was confusing and now explain the different data sets, and in which figures they can be found, in more detail in the revised manuscript. We also adapted the figure legend to be more clear (page 9).

Reviewer #2 (Remarks to the Author):

Papagiannoula and co-workers report on the interaction between Eps15 and Dab2 primarily applying high-resolution NMR spectroscopic approaches (analysis of chemical shift perturbations (1H, 15N), (15N based) spin relaxation R1, R1p, hNOE, relaxation dispersion). Schemes for isotopic labelling have been adapted such that potential interactions can be specifically followed on a residue-by-residue level. Samples have been also varied in the length of the primary sequence enabling a piece-by-piece interpretation of data acquired. A rather small part is provided by the application of fluorescence microscopy.

This study clearly profits from the in-depth NMR spectroscopic analysis of the interaction network existing between Eps15 and Dab2. This is a major strength of this study. The presentation of the data is straightforward and done with care.

However, capturing the width of the interaction network that is existing between Eps15 and Dab2 is challenging. Especially when it comes to an appropriate presentation of the results and interpretation for the broadly interested readership of Nature Communications. To be honest, it has been tough to digest all the information provided in this manuscript. Thus, it has not clear to me why a specific experiment has been conducted as it has been. The authors are encouraged to better motivate the procedures done. Please do not get me wrong, the authors worked hard. However, the sheer number of experimental data does not directly correlate with the impact of the study conducted. It should be clearly explained why a combination of specific methods is used and is appropriate to answer the underlying scientific question. This holds especially true when the study shall be published in a well-recognized, multidisciplinary journal.

We thank the reviewer for this positive assessment of our data and for encouraging us to explain our motivation to do certain experiments better. We now present a revised manuscript in which we critically reviewed the text describing our experiments, the rationale behind choosing what experiments to run and how to interpret experiments. We feel that this thorough revision significantly strengthened our manuscript and the conclusions taken, and makes it more accessible for readers with little or no expertise in NMR spectroscopy.

We have now also included a more detailed analysis of the Eps15 phase separation and co-phase separation with Dab2₃₂₀₋₄₉₅ as proposed by reviewer 1. These data are very much in line with our NMR experiments and we are convinced that this extension of the multi-disciplinary nature of our manuscript further broadens its scope and makes it interesting to readers from different disciplines.

To go along these lines, the idea behind Figure 7 is great. Unfortunately, this Figure is very hard to capture. Color coding, marking of phenylalanines, NPF and other motifs included, ... , is consistent with Figures presented in this manuscript before. But the key message that the authors aim to address (interaction network between Dab2 and Eps15) got somewhere lost by presenting all the details ...

We thank the reviewer for pointing out that our former Fig. 7 was partially confusing. To make the figure easier to read and to explain in the text, we now extended it into three panels (see also PbyP Fig. 5).

Panel A contains a lot of detail, but a very simplified visualization of Dab2₃₂₀₋₄₉₅ and Eps15, showing the detected interactions between the different proteins. We have now also pointed out the precise interaction regions within Eps15's IDR, which was absent in the previous figure.

In panel B, we now illustrate how the interactions may look like in space. In this figure we maintain the illustration of the different motifs, since our aim is to illustrate which molecular interactions are the reason for the overall interaction between the two proteins.

Panel C shows how an organization within a liquid-liquid phase separated state might look like. We agree with the reviewer that the previous display of the phase separated state was very small and contained a lot of detail. So we increased the figure size of the droplet illustration and removed, in this part of the figure, the molecular interaction sites. Instead, we have illustrated inter-molecular interactions between Eps15 and Dab2₃₂₀₋₄₉₅ in red shading and intra-molecular interactions between Eps15 EH domains and Eps15's IDR in pink shading.

PbyP Figure 5: Schematic of Eps15 and Dab2₃₂₀₋₄₉₅ interactions and phase separation. (A) Illustration of the complex interaction network between Eps15 and Dab2₃₂₀₋₄₉₅ including intra-molecular interactions within Eps15. Interactions are visualized with red dashed lines. **(B)** Cartoon of self-interactions within Eps15 and of interactions between Eps15 and Dab2₃₂₀₋₄₉₅. Eps15's EH domains interact with Eps15IDR and with Dab2₃₂₀₋₄₉₅. EH2 and EH3 tumble together. All three EH domains interact with the NPF motif within Dab2₃₂₀₋₄₉₅, while EH2 and EH3 are promiscuous binders. **(C)** The weak multivalent interaction network of the two IDR's with EH123 likely contribute to co-phase separation of Eps15 and Dab2₃₂₀₋₄₉₅. Inter-molecular interactions are highlighted in red shading and intra-molecular interactions are highlighted in pink shading.

Other comments:

(i) SF1: Which functional consequences do the multiple resonance signals possess?

The authors also speculate about the potential presence of cis/trans isomerization. Can this be (experimentally) confirmed?

The Dab₂₃₂₀₋₄₉₅ protein construct contains 23 proline residues. While cis-proline conformations are generally rare in folded proteins, they can be more prevalent in intrinsically disordered proteins. The population of cis conformers depends on the local environment of each proline and the overall flexibility of the protein chain. As a result, it is difficult to predict in advance how many cis-proline conformations will be significantly populated, or how many neighboring residues will exhibit distinct chemical shifts due to these states. However, if a ¹H-¹⁵N HSQC spectrum reveals more peaks than the number of non-proline residues in the sequence, this can indicate the presence of multiple conformational states in slow exchange —likely arising from cis-trans isomerization of prolines. This interpretation is further supported by backbone assignment spectra, which unambiguously link each peak to its position in the protein sequence, allowing the identification of duplicated peaks corresponding to residues adjacent to prolines. While this approach does not reveal which peaks belong to the cis or trans state, it confirms that both are populated.

This is also the case for some residues in Dab₂₃₂₀₋₄₉₅. Particularly evident is the population of both trans and cis proline states on the example of the only tryptophan in the protein chain, which has two prolines as neighbors within the sequence. The tryptophan side chain is clearly distinguished within the spectrum and shows indeed four peaks, not one as expected based on the NH-bond in its side chain. The different peaks stem from the neighboring prolines sampling both cis and trans states, leading to distinct tryptophan side chain peaks for trans-trans, trans-cis, cis-trans and cis-cis conformations of the neighboring prolines (PbyP Fig. 6). This behavior is furthermore maintained in the smaller Dab₂₃₅₈₋₃₉₀ which encompasses the single tryptophan and was used for assignment purposes.

PbyP Figure 6: Cis-trans proline isomerization on the example of a tryptophan flanked by two prolines. (A) Chemical structure of tryptophan, showing the two nitrogens bound to proton in blue font. One one NH-bond is in the side chain. Structure was created with BioRender.com. **(B)** Zoom into a ¹H-¹⁵N HSQC spectrum showing the peaks corresponding to tryptophan side chains, measured with Dab₂₃₂₀₋₄₉₅. Shown are the four peaks corresponding to the isoforms cis and trans of the preceding and following proline residues, which affect the chemical shift of the NH side chain peak of tryptophan.

In our manuscript, we mention cis-trans isomerization only related to Dab₂₃₂₀₋₄₉₅ assignment in Supplementary Fig. 1, because it explains additional peaks in the spectrum. However, we did not see cis-trans isomerization of prolines playing a role in the interactions we studied in this project. And the interaction between Dab₂₃₂₀₋₄₉₅ and Eps15 EH domains, appears not to have functional consequences on the isomerization. For clarity, we always show the major form in the figures. This statement is now also included in the revised manuscript (page 16).

There are a couple of resonance signals present in the corresponding HSQC spectrum that are not labeled. To my mind one should clearly label them as e.g. “not assigned” are similar.

We thank the reviewer for this suggestion. To clearly label the unassigned peaks we have added “*” next to those unassigned peaks that had comparable signal intensity as compared to the assigned peaks in all relevant supplementary figures. Furthermore, the assignments of Dab2₃₂₀₋₄₉₅ and Eps15_{IDR} can be accessed in the BMRB providing the exact position of assigned resonances (accession numbers 52613 (Dab2₃₂₀₋₄₉₅), 52866 (Eps15_{IDR} 481-581), 52864 (Eps15_{IDR} 569-671), 52863 (Eps15_{IDR} 648-780), and 52867 (Eps15_{IDR} 761-896)). The assignments of EH1, EH2, and EH3 in our experimental conditions were not uploaded to BMRB since assignments of these domains already exist (see for example BMRB accession numbers 4140 (EH1), 4288 (EH2), 4381 (EH3)).

Some spectra present tryptophan side chains, which is now also clearly indicated in the respective figure legends.

(ii) Have the authors thought about the acquisition of hNOE data in complex state(s)? Potentially, data shown in SF4 C may act as a good starting point.

We thank the reviewer for bringing up the idea of measuring hetNOE data in the complex state(s). While the R_{1ρ} experiments, which we routinely measured at different titration steps, is sensitive to segmental motion within the IDP and rotational tumbling affected by the binding of the much larger, and folded, interaction partners, hetNOE is sensitive to much faster, internal, motion.^{4,5}

The effect of complex formation on internal motion of the IDP would definitely be very interesting. The fully bound state of both Dab2₃₂₀₋₄₉₅ and Eps15_{IDR} is, however, hard to access due to dramatically increased line broadening upon complex formation. The interacting residues are invisible in the fully formed complex, also testified by the strongly increased R_{1ρ} in the course of the titration and before saturation is reached.

We thus decided to try to access an intermediate titration step of 10% EH123 and EH2 added to ¹⁵N Dab2₃₂₀₋₄₉₅, at which the main binding site, the NPF motif, already starts binding, but the corresponding peaks remain intense (see also Fig. 1). Under these conditions, no difference in the hetNOE can be observed (PbyP Fig. 7A). We therefore recorded an additional hetNOE of ¹⁵N Dab2₃₂₀₋₄₉₅ with 100% EH123. However, aside from just around the motifs, there was still no difference in the hetNOEs (PbyP Fig 7B).

Even though ¹H-¹⁵N hetNOEs do reflect changes in internal motion within an IDP chain,^{4,6} hetNOEs are actually not often measured upon complex formation. For a folded protein, for example, bound to an IDP, hetNOE values do not change much despite clear differences in R₁ and R₂.⁷ With interaction sites in Dab2₃₂₀₋₄₉₅ of only few residues in length, intermediate to fast exchange between the bound and the unbound state on the chemical shift and relaxation time scale throughout most regions of the proteins, and very weak binding affinities between Dab2₃₂₀₋₄₉₅ and its EH domain partners, we indeed expect the internal motions to only be minimally perturbed and hetNOE values essentially not affected.

PbyP Figure 7: ^1H - ^{15}N heteronuclear Overhauser effects (hetNOEs) of $\text{Dab2}_{320-495}$ alone and in complex. Shown are values measured at a ^1H frequency of 600 MHz of 100 μM $\text{Dab2}_{320-495}$ in the absence and in the presence of 10 μM EH2 or EH123 (top) and comparing the presence of 10 μM and 100 μM EH123 (bottom). The color legend is indicated in the figure.

(iii) SF7: Presenting results for fitting of experimental data is not appropriately done. The authors may provide a table or similar for the results they obtained while conducting fitting of data to an appropriate model.

We agree that the results for fitting our experimental data will be more clearly presented in the form of a table, and thank the reviewer for this suggestion. We have now included tables in the supplementary material along with the figures of the fits (Supplementary Tables 1 and 2, and Supplementary Fig. 7). We have chosen to keep these data in the supplement, acknowledging that the K_D fits are associated with quite large errors mainly due to difficulties in obtaining (visible) peaks of saturated states. The fitting model can be found in the methods section (page 16).

We also show the K_D values fitted from chemical shifts (PbyP Table 1) and from CPMG relaxation dispersion (PbyP Table 2) in this response.

Dab2 residue number	Kd value from fit to CSPs			
	EH1 (mM)	EH2 (mM)	EH3 (mM)	EH123 (μM)
341	n.d.	1.2 ± 0.6	1.2 ± 0.3	218 ± 14
342	n.d.	1 ± 0.6	2.6 ± 0.3	191 ± 60
366	n.d.	4 ± 3	4 ± 2	160 ± 5
400	0.35 ± 0.05	0.340 ± 0.09	1.39 ± 0.09	104 ± 4
458	n.d.	n.d.	2 ± 1	217 ± 26
460	n.d.	4 ± 3	4 ± 1	253 ± 10
484	n.d.	3 ± 3	4 ± 1	291 ± 7
486	n.d.	10 ± 10	3.0 ± 0.9	378 ± 16

PbyP Table 1: K_D values fitted for $\text{Dab2}_{320-495}$ binding to the different EH domains based on chemical shift perturbations. In this table, some interactions either do not exist (non-NFP interactions between

Dab2₃₂₀₋₄₉₅ and EH1 for example) or have too weak affinities to be fitted. Those are marked with not determined (n.d.). The data corresponding to the fits presented in Supplementary Fig. 7A.

Sample	Parameters from CPMG fit		Calculated K_D (μM)
	k_{ex} (s^{-1})	Percentage bound (%)	
100 μM 15N Dab2 ₃₂₀₋₄₉₅ +10 μM EH2	149 \pm 14	3.3 \pm 0.3	196

PbyP Table 2: K_D values fitted for Dab2₃₂₀₋₄₉₅ binding to EH2 assessed by CPMG relaxation dispersion. Shown are the exchange rate (k_{ex}), the percentage bound, and the K_D value calculated from the percentage bound and the protein concentrations used in the experiment (see also Supplementary Fig. 7B).

(iv) SF10: There is a resonance signal arising at about (10.2 / 107) ppm. The authors are encouraged to provide more information on it (assignment possible?). Moreover: This spectrum has been acquired in TROSY mode. I guess it is an HSQC, isn't it? The authors are encouraged to adapt the legend that accompanies this Figure. There are a couple of cross peaks arising that are not assigned. The authors are encouraged to provide crucial information on it.

We think the reviewer refers to the peak observed in Supplementary Fig. 9 (assignment of EH2), and Supplementary Fig. 10 in point (v). The peak arising about (10.2 / 107) ppm corresponds to a tryptophan side chain that is aliased due to the ^{15}N sweep width chosen in the experiment. To illustrate this, we re-recorded the same experiment with three different sweep widths, showing that the true position of the peaks lies well within the chemical shift range expected for the NH-bond in the tryptophan side chain (PbyP Fig. 8). We now explain the nature of that peak in the figure legend.

PbyP Figure 8: ^1H - ^{15}N TROSY-HSQC spectrum of EH2 recorded with different ^{15}N sweep widths. This experiment shows the original position of the tryptophan side chain peak (around 10.2 ppm ^1H chemical shift) as well as an aliased position. The different positions of this peak, as well as another peak aliased in the spectra, are illustrated by red arrows.

The spectra of EH domains recorded in TROSY mode to benefit from improved relaxation properties of the folded proteins are indeed ^1H - ^{15}N correlation experiments. To avoid ambiguity, we now clarify in the revised manuscript that the spectra are ^1H - ^{15}N TROSY HSQC experiments.

(v) SF11/SF11: See comments made to SF10.

Supplementary Fig. 10 shows the assignment of EH3. The tryptophan side chain is at its correct chemical shift in this figure. We now point out that this peak is a tryptophan side chain in the figure legend.

As in Supplementary Fig. 9, we now include the information that the spectrum is a ^1H - ^{15}N TROSY HSQC.

(vi) The text of the manuscript reads rather qualitative. Sadly, there are very few numbers mentioned even experimentally determined (e.g. the manuscript is accompanied by a Supplementary Information that comprises 21 (!) Figures). In my opinion, unraveling an interaction network spanned among molecules profits a lot by adding quantities. The authors are strongly encouraged to work on it.

We thank the reviewer for pointing this out. The interactions between the different partners in this interaction network (EH domains with Dab2₃₂₀₋₄₉₅ and Eps15_{IDR}) are very weak, even compared to other interactions that are commonly measured using NMR spectroscopy.^{6,8,9} This has as a consequence, that the saturated state can not be reached and, experimentally, we only have access to titration steps under which only a small percentage of the IDR is bound.

We have nevertheless quantified the affinities between Dab2₃₂₀₋₄₉₅ and the different EH domains, either using chemical shift titrations or using CPMG relaxation dispersion for the interaction with EH2 and Dab2₃₂₀₋₄₉₅ (Supplementary Fig. 7). In the original manuscript, we had printed the K_D values into the respective titration plots and in the figure legend for the CPMG experiment, respectively. We agree that these values were not very well visible and therefore now included two tables (Supplementary Tables 1 and 2, also PbyP Table 1 and Table 2), presenting relevant affinities obtained between Dab2₃₂₀₋₄₉₅ and Eps15 EH domains.

Due to the very weak interactions, the error bars obtained are quite large and we thus refrain from over-interpreting the results. However, we now do point out the order of interaction strengths more clearly in the revised manuscript (pages 5, 6, 11, and 12).

Interaction strengths between Eps15_{IDR} and EH domains have not been specifically quantified, since they correspond to the same kind of promiscuous interactions seen in Dab2₃₂₀₋₄₉₅ and the competition experiments undertaken suggest, that they are also of comparable strength. We now also discuss this aspect more clearly in the revised manuscript (page 10).

(vii) I am fully aware about the wealth of experimental data the authors already provide in the manuscript. While reading and evaluating the manuscript I have thought about potential NMR experiments that may facilitate the identification of the interaction network existing between Eps15 and Dab2. I would like to share these thoughts with the authors:

We thank the reviewer for bringing up these ideas for additional experiments to characterize the interaction between the different partners in the interaction network.

(a) Intermolecular NOEs - Direct structural readout (even a proof) of an existing interaction

Inter-molecular NOEs can indeed be highly beneficial for the analysis of a well-defined and stable protein complex and have even been used to characterize the interactions between EH domains and short peptides of IDPs,^{10,11} providing a detailed understanding about how EH domains can recognize their targets and contributing towards the understanding of NPF binding by EH domains. The Stonin2 peptide, which has led to 400 inter-molecular NOEs, has an affinity towards EH2 in the nanomolar range, thus orders of magnitudes stronger than the interactions we observed.¹⁰ An Hrb peptide comprising an NPF motif bound to EH2 led only to up to 64 inter-molecular NOEs with an affinity of 12 μ M, still much stronger than most interactions we observe.¹¹

While the work we present benefits from the studies describing precise binding poses between a short IDR peptide and EH2, this was not the goal of our study. Our work focuses on much longer protein segments with many, relatively low affinity interaction sites available on the IDRs to bind to EH domains. This illustrates how small binding motifs behave in the context of a much larger protein chain – an aspect that is to date still severely understudied. Consequently, the overall complex the proteins form is highly dynamic and interactions so transient that inter-molecular NOEs are unlikely to yield specific binding poses. Rather than focusing on discrete binding poses, our study aims to capture the dynamic and distributed nature of these interactions along the IDRs — an aspect for which inter-molecular NOEs unfortunately offer limited information.

(b) PREs (readout can be done using e.g. ¹H, ¹H-¹⁵N or even ¹⁹F resonance signals) - Strategically placed probes will thereby enable the determination of interacting nuclei that are further away than 6 Ångström (limit for 'traditional' homonuclear ¹H-¹H NOEs)

We highly appreciate the reviewer's suggestion to include PREs or ¹⁹F NMR to characterize the complexes formed. We are indeed extremely interested in understanding the long-range interactions within those highly dynamic complexes and we are very confident that PREs would pick up on the transient interactions taking place.

We have indeed an ongoing project, where we aim to use PREs as well as single molecule Förster resonance energy transfer (FRET), which is sensitive to even longer interactions up to 10 nanometers,¹² to assess the arrangement of the different EH domains with respect to each other and how they bind to IDPs. Combined with modeling approaches, which we are currently undertaking, this promises a three dimensional description of the involved proteins and their complexes in the context of a conformational ensemble. However, we feel that this is an entirely independent project towards a structural description of the complexes, which goes beyond the scope of the presented manuscript.

To my mind, suggestions (a) and (b) significantly expands the experimental repertoire beside acquiring data determining the relaxation in the rotating frame (R1 ρ) or conducting titration experiments (CSPs) as mainly done in the present manuscript.

We thank the reviewer again for thinking beyond the experiments we have presented in our manuscript. Without doubt, there are many interesting experiments that could provide exciting new insights into such complex interaction networks as described here, and we are confident that we or others will touch some of them in the future.

When we designed this study, we were really interested in how large IDRs, not just small peptides, interacted with their folded binding partners in clathrin mediated endocytosis. Which known and

unknown binding motifs played a role, what the affinities were, and how this could be important in the context of clathrin mediated endocytosis and phase separation initiating this process. We chose the techniques used as to best address our research question and we show unambiguous and novel insights into the interaction network between Dab2 and Eps15. Our data further answers, which residues in Eps15_{IDR} actually interact with EH domains to form liquid-like droplets. Our work provides a new view into the degree of multivalency and competition at the endocytic pit and nurtures the notion that both NPF and non-NPF interactions contribute to shaping this interaction network.

We would like to clarify that our study draws on a broad set of NMR and complementary methods. In addition to the $R_{1\rho}$ and CSP data emphasized in the main text, we performed extensive backbone assignments for eight different protein constructs (including the EH domains), analyzed secondary structure propensities using carbon chemical shifts, and conducted CPMG relaxation dispersion experiments to characterize affinities and exchange processes involving Dab2₃₂₀₋₅₉₄ and EH2. We also collected R_1 and hetNOE data to further examine protein dynamics.

Beyond NMR, we have also integrated complementary techniques, including fluorescence imaging — particularly during the revision phase — to assess the formation and concentration dependence of Eps15 and Eps15:Dab2₃₂₀₋₄₉₅ condensates. We hope this clarifies that the study is based on a comprehensive and multi-disciplinary experimental approach, with NMR $R_{1\rho}$ and CSP experiments playing a central but not exclusive role.

Minor:

All pages of the manuscript possess page number “50”.

We thank the reviewer for pointing this out. We have implemented the correct page numbers in the revised manuscript.

Reviewer #3 (Remarks to the Author):

This exciting manuscript from Papagiannoula and colleagues provides a detailed glimpse into the complexity of the dynamic, multivalent interaction landscape of two intrinsically disordered proteins involved in clathrin-mediated endocytosis, Eps15 and Dab2. Using NMR spectroscopy, which is exceptionally well-suited for characterizing these highly dynamic disordered proteins, the authors demonstrate the importance of phenylalanine residues both within and outside of previously identified binding motifs in mediating intermolecular (Eps15-Dab2 and Eps15-Eps15) interactions as well as facilitating intramolecular interactions between the Eps15 IDR and the EH domains. The data are presented clearly and logically and thoroughly support the authors' conclusions. This work expands current knowledge of protein interactions in the early stages of endocytosis and will be of great interest to many.

I have only a few minor questions and comments that I hope can be addressed to provide additional clarity on the important concepts and data presented in the manuscript:

We thank the reviewer for the very positive assessment of our work and its impact.

1. The experiments comparing the interactions of Dab2 with the isolated EH domains and the EH123 construct clearly demonstrate that there is benefit to having 3 of these domains linked, presumably due to avidity effects that enhance the binding affinity. In the competition experiments, the data show that the Eps15 IDR can partially outcompete Dab2 for binding to the EH domains when added in trans...but might this competition be more effective in the context of full-length Eps15? It is easy to envision that interactions of the Eps15 IDR with the EH domains would be even more favorable in the context of the full-length protein. How do the authors think this would impact binding to Dab2 and other interaction partners? Some further discussion of how the intramolecular interactions of Eps15 may differ in the intact protein versus in mixtures of truncated forms is warranted.

We thank the reviewer for encouraging us to interpret our data more in the context of the full length proteins, which will also be relevant for their function in clathrin-mediated endocytosis. We indeed think that the interactions between Eps15 EH domains and its IDR will benefit from their close proximity in the context of a full length construct. They can likely occur intra- and inter-molecularly, thus leading to liquid-liquid phase separation. We have now included a discussion on proximity in the revised manuscript page 13.

Thanks to our new and very extensive study of liquid-liquid phase separation by Eps15 and Dab2₃₂₀₋₄₉₅ (PbyP Fig. 1-4, as well as Fig. 6 and Supplementary Fig. 22 of the revised manuscript), we now know that Dab2₃₂₀₋₄₉₅ does not inhibit Eps15 phase separation over a range of concentration, suggesting that the competitive effect of this small construct of Dab2 might not be strong enough.

However, we do think that this can be different at the endocytic pit, where full length Dab2 and other NPF-containing proteins are present. As explained in the reply to reviewer 1, Eps15 condensates are thought to initiate clathrin-mediated endocytosis¹ leading to accumulation of downstream client proteins that make endocytosis progress. When other CLASPs/accessory proteins enriched at the pit, Eps15 moves away.^{2,3} A much higher concentration of competitive interaction partners from different regions of Dab2, but also other interaction partners may thus be needed to affect phase separation or segregate these proteins out of Eps15 condensates. We have now included this discussion into the revised manuscript (page 14).

2. I am curious if the authors have attempted to determine the binding affinities by any method other than NMR. Might there be tighter interactions that aren't quantifiable by NMR (ie. for resonances that are broadened beyond detection)?

We have indeed try to do isothermal titration calorimetry (ITC) of Dab2₃₂₀₋₄₉₅ and its strongest binding partner among the Eps15 domains: EH123. However, we could not observe any binding, either because the overall affinity, with contributions from all the different individual interactions sites, remains too weak, or because the interaction is not enthalpic. The results of the experiment are shown in PbyP Fig. 9.

PbyP Figure 9: ITC of Dab₃₂₀₋₄₉₅ with EH123. 100 μM EH123 was placed in the cell for this experiment, 819 μM Dab₃₂₀₋₄₉₅ was placed in the syringe. The upper plot shows the differential power (DP) at the different injections over time. The lower plot shows the corresponding differences in enthalpy (ΔH) plotted against the molar ratio of the two proteins (Dab₃₂₀₋₄₉₅/EH123).

We have indeed used NMR chemical shift perturbations during the titration to get an estimate of affinities along all residues. For this, we have followed the peaks until disappearance. While peak broadening prevents us from observing the titration until saturation, the position of the peaks were used to fit a K_D value according to a two-state model. Due to the weak affinities and the partially few points accessible to the fit, the error bars obtained are relatively large. Nevertheless, they allow us to judge the overall range of affinities within the complex. For those residues that showed exchange line broadening in the microsecond to millisecond range, we also performed CPMG relaxation dispersion and fitted the data, again using a two-site exchange model. The K_D extracted from this fit is in overall good agreement with the K_D values extracted from fitting the chemical shift

displacements. These data, shown in Supplementary Fig. 7, are now also presented in two tables (Supplementary Tables 1 and 2) for more clarity.

While the error bars on the K_D values that we determined are rather large, we do have a good estimate on the order of affinity and can thus exclude that much stronger interactions escape our analysis. We now discuss the different affinities and their interpretation in more detail in the revised manuscript (pages 5, 6, 10, 11, and 12).

3. The text labels in many of the figures are far too tiny to be legible, even at high magnification. This is particularly true for Figure 4C, Figure S6, and Figure S7, but larger labels would generally be helpful throughout.

We thank the reviewer for pointing this out. We have increased the font size throughout the figures and hope that they are now well legible.

1. Day, K. J. *et al.* Liquid-like protein interactions catalyse assembly of endocytic vesicles. *Nat Cell Biol* **23**, 366–376 (2021).
2. Sochacki, K. A., Dickey, A. M., Strub, M.-P. & Taraska, J. W. Endocytic proteins are partitioned at the edge of the clathrin lattice in mammalian cells. *Nat Cell Biol* **19**, 352–361 (2017).
3. Taylor, M. J., Perrais, D. & Merrifield, C. J. A High Precision Survey of the Molecular Dynamics of Mammalian Clathrin-Mediated Endocytosis. *PLOS Biology* **9**, e1000604 (2011).
4. Abyzov, A. *et al.* Identification of Dynamic Modes in an Intrinsically Disordered Protein Using Temperature-Dependent NMR Relaxation. *J Am Chem Soc* **138**, 6240–6251 (2016).
5. Salvi, N., Abyzov, A. & Blackledge, M. Solvent-dependent segmental dynamics in intrinsically disordered proteins. *Sci Adv* **5**, eaax2348 (2019).
6. Milles, S. *et al.* An ultraweak interaction in the intrinsically disordered replication machinery is essential for measles virus function. *Sci Adv* **4**, eaat7778 (2018).
7. Bugge, K. *et al.* Role of charges in a dynamic disordered complex between an IDP and a folded domain. *Nat Commun* **16**, 3242 (2025).
8. Naudi-Fabra, S. *et al.* An extended interaction site determines binding between AP180 and AP2 in clathrin mediated endocytosis. *Nat Commun* **15**, 5884 (2024).
9. Guseva, S. *et al.* Measles virus nucleo- and phosphoproteins form liquid-like phase-separated compartments that promote nucleocapsid assembly. *Sci Adv* **6**, eaaz7095 (2020).
10. Rumpf, J. *et al.* Structure of the Eps15-stonin2 complex provides a molecular explanation for EH-domain ligand specificity. *EMBO J* **27**, 558–569 (2008).
11. de Beer, T. *et al.* Molecular mechanism of NPF recognition by EH domains. *Nat Struct Biol* **7**, 1018–1022 (2000).
12. Naudi-Fabra, S., Tengo, M., Jensen, M. R., Blackledge, M. & Milles, S. Quantitative Description of Intrinsically Disordered Proteins Using Single-Molecule FRET, NMR, and SAXS. *J Am Chem Soc* **143**, 20109–20121 (2021).

Papagiannoula and co-workers report on the interaction between Eps15 and Dab2 primarily applying high-resolution NMR spectroscopic approaches (analysis of chemical shift perturbations (^1H , ^{15}N), (^{15}N based) spin relaxation R_1 , $R_{1\rho}$, hNOE, relaxation dispersion). Schemes for isotopic labelling have been adapted such that potential interactions can be specifically followed on a residue-by-residue level. Samples have been also varied in the length of the primary sequence enabling a piece-by-piece interpretation of data acquired. A rather small part is provided by the application of fluorescence microscopy.

This study clearly profits from the in-depth NMR spectroscopic analysis of the interaction network existing between Eps15 and Dab2. This is a major strength of this study. The presentation of the data is straightforward and done with care.

However, capturing the width of the interaction network that is existing between Eps15 and Dab2 is challenging. Especially when it comes to an appropriate presentation of the results and interpretation for the broadly interested readership of Nature Communications. To be honest, it has been tough to digest all the information provided in this manuscript. Thus, it has not clear to me why a specific experiment has been conducted as it has been. The authors are encouraged to better motivate the procedures done. Please do not get me wrong, the authors worked hard. However, the sheer number of experimental data does not directly correlate with the impact of the study conducted. It should be clearly explained why a combination of specific methods is used and is appropriate to answer the underlying scientific question. This holds especially true when the study shall be published in a well-recognized, multidisciplinary journal.

To go along these lines, the idea behind Figure 7 is great. Unfortunately, this Figure is very hard to capture. Color coding, marking of phenylalanines, NPF and other motifs included, ... , is consistent with Figures presented in this manuscript before. But the key message that the authors aim to address (interaction network between Dab2 and Eps15) got somewhere lost by presenting all the details ...

Other comments:

(i) SF1: Which functional consequences do the multiple resonance signals possess?

The authors also speculate about the potential presence of cis/trans isomerization. Can this be (experimentally) confirmed?

There are a couple of resonance signals present in the corresponding HSQC spectrum that are not labeled. To my mind one should clearly label them as e.g. "not assigned" are similar.

(ii) Have the authors thought about the acquisition of hNOE data in complex state(s)? Potentially, data shown in SF4 C may act as a good starting point.

(iii) SF7: Presenting results for fitting of experimental data is not appropriately done. The authors may provide a table or similar for the results they obtained while conducting fitting of data to an appropriate model.

(iv) SF10: There is a resonance signal arising at about (10.2 / 107) ppm. The authors are encouraged to provide more information on it (assignment possible?). Moreover: This

spectrum has been acquired in TROSY mode. I guess it is an HSQC, isn't it? The authors are encouraged to adapt the legend that accompanies this Figure. There are a couple of cross peaks arising that are not assigned. The authors are encouraged to provide crucial information on it.

(v) SF11/SF11: See comments made to SF10.

(vi) The text of the manuscript reads rather qualitative. Sadly, there are very few numbers mentioned even experimentally determined (e.g. the manuscript is accompanied by a Supplementary Information that comprises 21 (!) Figures). In my opinion, unraveling an interaction network spanned among molecules profits a lot by adding quantities. The authors are strongly encouraged to work on it.

(vii) I am fully aware about the wealth of experimental data the authors already provide in the manuscript. While reading and evaluating the manuscript I have thought about potential NMR experiments that may facilitate the identification of the interaction network existing between Eps15 and Dab2. I would like to share these thoughts with the authors:

(a) Intermolecular NOEs - Direct structural readout (even a proof) of an existing interaction

(b) PREs (readout can be done using e.g. ^1H , ^1H - ^{15}N or even ^{19}F resonance signals) - Strategically placed probes will thereby enable the determination of interacting nuclei that are further away than 6 Ångström (limit for 'traditional' homonuclear ^1H - ^1H NOEs)

To my mind, suggestions (a) and (b) significantly expands the experimental repertoire beside acquiring data determining the relaxation in the rotating frame ($R_{1\rho}$) or conducting titration experiments (CSPs) as mainly done in the present manuscript.

Minor:

All pages of the manuscript possess page number "50".